# Adversarial Training for Graph Neural Networks: Pitfalls, Solutions, and New Directions

**Lukas Gosch**[1*], **Simon Geisler**[1*], **Daniel Sturm**[1*], **Bertrand Charpentier**[1],
**Daniel Zügner**[1,2], **Stephan Günnemann**[1]
[1]Department of Computer Science & Munich Data Science Institute
Technical University of Munich
[2]Microsoft Research
{l.gosch, s.geisler, da.sturm, s.guennemann}@tum.de | dzuegner@microsoft.com

## Abstract

Despite its success in the image domain, adversarial training did not (yet) stand out as an effective defense for Graph Neural Networks (GNNs) against graph structure perturbations. In the pursuit of fixing adversarial training *(1)* we show and overcome fundamental theoretical as well as practical limitations of the adopted graph learning setting in prior work; *(2)* we reveal that flexible GNNs based on learnable graph diffusion are able to adjust to adversarial perturbations, while the learned message passing scheme is naturally interpretable; *(3)* we introduce the first attack for structure perturbations that, while targeting multiple nodes at once, is capable of handling global (graph-level) as well as local (node-level) constraints. Including these contributions, we demonstrate that adversarial training is a state-of-the-art defense against adversarial structure perturbations.[1]

## 1 Introduction

Adversarial training has weathered the test of time and stands out as one of the few effective measures against adversarial perturbations. While this is particularly true for numerical input data like images [1], it is not yet established as an effective method to defend predictions of Graph Neural Networks (GNNs) against graph structure perturbations.

Although previous work reported some robustness gains by using adversarial training in GNNs [2, 3], closer inspection highlights two main shortcomings: (i) their learning setting leads to a biased evaluation and (ii) the studied architectures seem to struggle to learn robust representations.

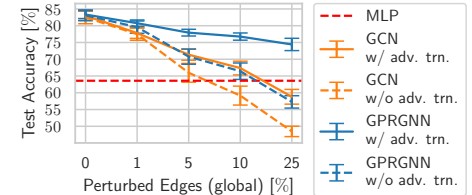

Figure 1: Robust diffusion (GPRGNN) vs. GCN on (inductive) Cora-ML. We report adversarial training and standard training.

*(i) Learning setting.* In the previously studied transductive learning settings (see Table 3), clean validation and test nodes are known during training. Thus, *perfect robustness* can be achieved by memorizing the training graph. This can lead to a false impression of robustness. Indeed, we find that the gains of the adversarial training approach from Xu et al. [2] largely stem from exploiting this flaw. Motivated by this finding, we revisit adversarial training for node classification under structure perturbations in a fully inductive setting (i.e., validation/test nodes are excluded during training). Thus, our results do not suffer from the same evaluation pitfall and pertain to a more challenging and realistic scenario.

---

[*]Equal contribution.
[1]Project page: https://www.cs.cit.tum.de/daml/adversarial-training/

37th Conference on Neural Information Processing Systems (NeurIPS 2023).

*(ii) GNN architectures.* The studied GNNs like GCN [4] or APPNP [5] all seem to share the same fate as they have a limited ability to adjust their message passing to counteract adversarial perturbations. Instead, we propose to use flexible message-passing schemes based on *learnable diffusion*. The main motivation behind this choice is the ability to approximate any spectral graph filter. Thus, adversarial training may choose the most robust filter that achieves a competitive training loss. Thereby, we significantly improve robustness compared to previous work, while yielding an interpretable message-passing scheme, and making the evaluation bias in the transductive setting (i) apparent.

*More realistic perturbations.* Inspecting robust diffusion indicates that previously studied perturbation sets [2, 3] are overly permissive and unrealistic. Prior work only constrained the *global* (graph-level) number of inserted and removed edges, despite studying node classification, where predictions are inherently *local* (node-level). As a result, commonly, an adversary has been allowed to add or remove a number of edges that exceeds the average node degree by 100 times or more, providing ample leeway for a complete change of many node neighborhoods through rewiring. E.g., on Cora-ML 5% edge-changes correspond to 798 changes, but the average degree is 5.68. To prevent such degenerate perturbations [6], we propose Locally constrained Randomized Block Coordinate Descent (LR-BCD), the first attack that, while targeting multiple nodes at once, is able to constrain *both* local perturbations per node and global ones. Even though local constraints are well-studied for attacks targeting a single node [7], surprisingly, there was no attack incorporating these for jointly attacking a set of nodes at once. Thus, LR-BCD fills an important gap in the literature, while being efficient and effective.

Addressing the aforementioned points, we substantially boost empirical and certifiable adversarial robustness up to the level of state-of-the-art defenses for GNNs. For example, in Figure 1, we show a 4-fold increased robustness over standard training (measured in decline in accuracy after an attack).

**Contributions. (1)** We show theoretically and empirically that in the transductive learning setting previously studied for adversarial training, one can trivially achieve perfect robustness. We show that the full inductive setting does not have this limitation and, consequently, revisit adversarial training in this setting (see Section 2). **(2)** By leveraging more flexible GNN architectures based on learnable diffusion, we significantly improve upon the robustness under adversarial training (see Section 3). **(3)** We implement more realistic adversaries for structure perturbations with a novel attack that constrains perturbations globally and locally for each node in the graph (see Section 4).

## 2 Learning Settings: Transductive vs. Inductive Adversarial Training

We first give background on adversarial training and self-training. Then, in Section 2.1, we discuss the transductive learning setting and its shortcomings in contrast to inductive learning in the context of robust generalization. Following previous work, we focus on node classification and assume we are given an undirected training graph $\mathcal{G} = (\mathbf{X}, \mathbf{A})$ consisting of $n$ nodes of which $m$ are labeled. By $\mathbf{A} \in \{0,1\}^{n \times n}$ we denote the adjacency matrix, by $\mathbf{X} \in \mathbb{R}^{n \times d}$ the feature matrix, and by $y_i \in \{1, ..., C\}$ the label of a node $i$, summarized for all nodes in the vector $\boldsymbol{y}$.

**Adversarial Training.** Adversarial training optimizes the parameters of a GNN $f_\theta$ on an adversarially perturbed input graph with the goal to increase robustness against test-time (evasion) attacks (i.e., attacks against the test graph after training). The objective during training is

$$\arg\min_\theta \max_{\tilde{\mathcal{G}} \in \mathcal{B}(\mathcal{G})} \sum_{i=1}^m \ell(f_\theta(\tilde{\mathcal{G}})_i, y_i) \tag{1}$$

where $f_\theta(\tilde{\mathcal{G}})_i$ is the GNN-prediction for node $i$ based on the perturbed graph $\tilde{\mathcal{G}}$, $\ell$ is a chosen loss function, and $\mathcal{B}(\mathcal{G})$ is the set of allowed perturbed graphs given the clean graph $\mathcal{G}$. As in previous work, we assume that an adversary maliciously changes the graph structure by inserting or deleting edges. Then, $\mathcal{B}(\mathcal{G})$ can be defined by restricting the total number of malicious edge changes in the graph to $\Delta \in \mathbb{N}_{\geq 0}$ (called a global constraint) and/or restricting the total number of edge changes in each neighborhood of a node $i$ to $\Delta_i^{(l)} \in \mathbb{N}_{\geq 0}$ (called local constraints, see Section 4). Solving Equation (1) is difficult in practice. Thus, we approximate it with alternating first-order optimization, i.e., we approximately solve Equation (1) by training $f_\theta$ not on the clean graph $\mathcal{G}$, but on a perturbed one $\tilde{\mathcal{G}}$ that is newly crafted in each iteration through attacking the current model (see Algorithm B.2).

**Self-training** is an established semi-supervised learning strategy [8] that leverages unlabeled nodes via pseudo-labeling and is applied by previous works on adversarial training for GNNs [2, 3]. For this, first

a model $f_{\theta'}$ is trained regularly using the $m$ labeled nodes, minimizing $\sum_{i=1}^{m} \ell(f_\theta(\mathcal{G})_i, y_i)$. Thereafter, a new (final) model $f_\theta$ is randomly initialized and trained on all nodes, using the known labels for the training nodes and pseudo-labels generated by $f_{\theta'}$ for the unlabeled nodes (see Algorithm B.3). Thus, if including self-training, Equation (1) changes to

$$\arg\min_{\theta} \max_{\tilde{\mathcal{G}} \in \mathcal{B}(\mathcal{G})} \left\{ \sum_{i=1}^{m} \ell(f_\theta(\tilde{\mathcal{G}})_i, y_i) + \sum_{i=m+1}^{n} \ell(f_\theta(\tilde{\mathcal{G}})_i, f_{\theta'}(\mathcal{G})_i) \right\} \qquad (2)$$

## 2.1 Transductive Setting

Transductive node classification is a common and well-studied graph learning problem [9, 4]. It aims to complete the node labeling of the given and partially labeled graph $\mathcal{G}$. More formally, the goal at test time is to accurately label the already known $n - m$ unlabeled nodes, i.e., to achieve minimal

$$\mathcal{L}_{0/1}(f_\theta) = \sum_{i=m+1}^{n} \ell_{0/1}(f_\theta(\mathcal{G})_i, y_i) \qquad (3)$$

This is in contrast to an inductive setting, where at test time a new (extended) graph $\mathcal{G}'$ (with labels $\boldsymbol{y}'$) is sampled conditioned on the training graph $\mathcal{G}$. Then, the goal is to optimally classify the newly sampled nodes $\mathcal{I}$ *in expectation* over possible $(\mathcal{G}', \boldsymbol{y}')$-pairs, i.e., to minimize $\mathbb{E}\left[\sum_{i \in \mathcal{I}} \ell_{0/1}(f_\theta(\mathcal{G}')_i, y_i')\right]$. That is, in transductive learning the $n-m$ unlabeled nodes known during training are considered test nodes, but in an inductive setting, new unseen test nodes are sampled. For additional details on the inductive setting, see Appendix A.

Many common graph benchmark datasets such as Cora, CiteSeer, or Pubmed [10, 11], are designed for transductive learning. Naturally, this setting has been adopted as a starting point for many works on robust graph learning [7, 2, 12], including all of the adversarial training literature (see Section 6). Here, the latter is concerned with defending against test-time (evasion) attacks in transductive node classification, i.e., it is assumed that after training an adversary can select a maliciously changed graph $\tilde{\mathcal{G}}$ out of the set of permissible perturbations $\mathcal{B}(\mathcal{G})$, with the goal to maximize misclassification:

$$\mathcal{L}_{adv}(f_\theta) = \max_{\tilde{\mathcal{G}} \in B(\mathcal{G})} \sum_{i=m+1}^{n} \ell_{0/1}(f_\theta(\tilde{\mathcal{G}})_i, y_i) \qquad (4)$$

Since the adversary changes the graph at test time (i.e., the changes are not known during training), this, strictly speaking, corresponds to an inductive setting [9], where the only change considered is adversarial. Now, the goal of *adversarial training* is to find parameters $\theta$ minimizing $\mathcal{L}_{adv}(f_\theta)$, corresponding to the optimal (robust) classifier under attack [1].

### 2.1.1 Theoretical Limitations

Defending against test-time (evasion) attacks in a transductive setting comes with conceptual limitations. In the case of graph learning, the test nodes are already known at training time and the only change is adversarial. Hence, we can design a defense algorithm $\mathcal{A}$ achieving *perfect robustness* without trading off accuracy through *memorizing* the (clean) training graph.

Formally, $\mathcal{A}$ takes a GNN $f_\theta$ as input and returns a slightly modified model $\tilde{f}_\theta$ corresponding to $f_\theta$ composed with a preprocessing routine (memory) that, during inference, replaces the perturbed graph $\tilde{\mathcal{G}} = (\tilde{\mathbf{X}}, \tilde{\mathbf{A}}) \in \mathcal{B}(\mathcal{G})$ with the clean graph $\mathcal{G} = (\mathbf{X}, \mathbf{A})$ known from training, resulting in $\tilde{f}_\theta(\tilde{\mathcal{G}}) = f_\theta(\mathcal{G})$. In other words, $\tilde{f}_\theta$ ignores every change to the graph including those of the adversary. Trivially, this results in the same (clean) misclassification rate $\mathcal{L}_{0/1}(f_\theta) = \mathcal{L}_{0/1}(\tilde{f}_\theta)$ because $\tilde{f}_\theta$ and $f_\theta$ have the same predictions on the clean graph, but also perfect robustness, in the sense of $\mathcal{L}_{0/1}(\tilde{f}_\theta) = \mathcal{L}_{adv}(\tilde{f}_\theta)$, as the predictions are not influenced by the adversary. Thus, we can state:

**Proposition 1.** *For transductive node classification, $\tilde{f}_\theta = \mathcal{A}(f_\theta)$ is a perfectly robust version of an arbitrary GNN $f_\theta$, in the sense of $\mathcal{L}_{0/1}(f_\theta) = \mathcal{L}_{0/1}(\tilde{f}_\theta) = \mathcal{L}_{adv}(\tilde{f}_\theta)$.*

However, what is probably most interesting about $\mathcal{A}$ is that we can use it to construct an *optimal solution* to the otherwise difficult saddle-point problem $\min_\theta \mathcal{L}_{adv}(f_\theta) = \min_\theta \max_{\tilde{\mathcal{G}} \in B(\mathcal{G})} \mathcal{L}_{0/1}(f_\theta)$ arising in adversarial training. Formally, we state (proof see Appendix C.1):

**Proposition 2.** *Assuming we solve the standard learning problem $\theta^* = \arg\min_\theta \mathcal{L}_{0/1}(f_\theta)$ and that $\mathcal{G} \in \mathcal{B}(\mathcal{G})$. Then, $\tilde{f}_{\theta^*} = \mathcal{A}(f_{\theta^*})$ is an optimal solution to the saddle-point problem arising in (transductive) adversarial training, in the sense of $\mathcal{L}_{adv}(\tilde{f}_{\theta^*}) = \min_\theta \mathcal{L}_{adv}(\tilde{f}_\theta) \leq \min_\theta \mathcal{L}_{adv}(f_\theta)$.*

In other words, at train time, from a GNN $f_{\theta^*}$ minimizing the clean error, we can construct a perfectly robust classifier $\tilde{f}_{\theta^*}$ minimizing the maximal misclassification rate under attack solely from memorizing the data available during training.

Note that in a fully inductive setting due to the expectation in the losses, neither Proposition 1 nor Proposition 2 hold, i.e., an optimal solution cannot be found through memorization (see Appendix A). Thus, inductive node classification does not suffer from the same theoretical limitations.

### 2.1.2 Empirical Limitations

Even though prior work did not achieve perfect robustness, we show below that the reported gains from adversarial training by Xu et al. [2] actually mostly stem from self-training, i.e., the "leakage" of information on the clean test nodes through pseudo-labeling. Then, we close the gap between theory and practice and show that perfect robustness, while preserving clean accuracy, can be achieved empirically through adversarial training – effectively solving the learning setting in prior work.

**Self-training is the (main) cause for robustness in transductive learning.** In Figure 2, we compare the (robust) test accuracies of a GCN using different training schemes: $(i)$ normal training, $(ii)$ adversarial training, $(iii)$ self-training, and $(iv)$ pairing self- and adversarial training. Indeed, Figure 2 shows that the main robustness gain actually stems from self-training and *not* adversarial training. Surprisingly, adversarial training without self-training can even *hurt* robustness, while adversarial training paired with self-training only slightly increases robust test accuracy compared to using self-training alone. These results are independent of the used architecture, dataset, or attack strength, as we show in Appendix C.2.2.

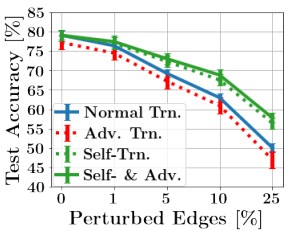
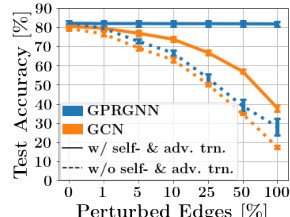

Figure 2: Robust test-accuracy of a GCN under different training schemes on Cora. Adversarial training uses a PGD-attack (10% pert. edges). Most robustness gains are due to self-training.

Figure 3: Adv. trained GPRGNN following Xu et al. [2] achieves perfect robustness on Cora. GCN baseline uses the same adv. training setup (training budget: 100% of edges).

**Adversarial training causes overfitting.** For transductive node classification, it is also empirically possible to achieve perfect robustness while maintaining clean test accuracy by combining self- with adversarial training and using a more flexible, learnable diffusion model (GPRGNN) as introduced in Section 3 In Figure 3, we show the test accuracy under severe perturbations for an adversarially trained GCN [2] and GPRGNN, both trained using a very strong adversary and pseudo-labels derived from self-training. The performance of the adversarially trained GCN reduces rapidly, while GPRGNN achieves (almost constant) perfect robustness. Since the pseudo-labels are derived from the clean graph, the clean graph is leaked in the training process. This allows GPRGNN to mimic the behavior of an MLP and overfit to the pseudo-labels, i.e., memorize the (node-feature, pseudo-label)-pairs it has seen during training. In other words, it finds a trivial solution to this learning setting, making its limitations evident similar to the theoretic solution discussed in Section 2.1.1. In Appendix C.2.1 we show that we achieve perfect robustness not only if using the PGD attack as Xu et al. [2], but also for many different attacks.

These findings question how to interpret the reported robustness gains of previous work, which all evaluate transductively and usually use self-training. Conceptually, being robust against test-time (evasion) attacks is most relevant if there can be a natural change in the existing data. Only then it is realistic to assume an adversary causing some of the change to be malicious and against which we want to be robust without losing our ability to generalize to new data. Therefore, in Section 5 we revisit adversarial training in an inductive setting, avoiding the same evaluation pitfalls. Indeed, we find that using robust diffusion (see Section 3), we are not only capable of solving transductive robustness, but learn to robustly generalize to unseen nodes far better than previously studied models.

## 3 Robust Diffusion: Combining Graph Diffusion with Adversarial Training

We propose to use learnable graph diffusion models able to approximate any graph filter in conjunction with adversarial training to obtain a *robust diffusion*. The key motivation is to use more flexible GNN architectures for adversarial training than previous studies. We not only show that a robust diffusion significantly outperforms other models used in previous work (see Section 5), but it also allows for increased *interpetability* as we can gain insights into the characteristics of such robustness-enhancing models from different perspectives.

**In fixed message passing** schemes of popular GNNs like GCN or APPNP, each layer can be interpreted as the *graph convolution* $g(\boldsymbol{\Lambda}) \circledast \boldsymbol{H} = \boldsymbol{V} g(\boldsymbol{\Lambda}) \boldsymbol{V}^\top \boldsymbol{H}$ between a *fixed* graph filter $g(\boldsymbol{\Lambda})$ and the transformed node attributes are $\boldsymbol{H} = f_\theta(\boldsymbol{X})$ using MLP $f_\theta$. This convolution is defined w.r.t. the (diagonalized) eigenvalues $\boldsymbol{\Lambda} \in \mathbb{R}^{n \times n}$ and eigenvectors $\boldsymbol{V} \in \mathbb{R}^{n \times n}$ of the Laplacian $\boldsymbol{L}' = \boldsymbol{I} - \boldsymbol{D}^{-1/2} \boldsymbol{A} \boldsymbol{D}^{-1/2}$ with diagonal degree matrix $\boldsymbol{D}$. Following the common reparametrization, instead of $\boldsymbol{L}'$, we use the "normalized adjacency" $\boldsymbol{L} = \boldsymbol{D}^{-1/2} \boldsymbol{A} \boldsymbol{D}^{-1/2}$ or, depending on the specific model, $\mathring{\boldsymbol{L}} = \mathring{\boldsymbol{D}}^{-1/2} \mathring{\boldsymbol{A}} \mathring{\boldsymbol{D}}^{-1/2}$ where $\mathring{\boldsymbol{A}} = \boldsymbol{A} + \boldsymbol{I}$ with node degrees $\mathring{\boldsymbol{D}}$. Then, many spatial GNNs relate to the $K$-order polynomial approximation $\boldsymbol{V} g(\boldsymbol{\Lambda}) \boldsymbol{V}^\top \boldsymbol{H} \approx \sum_{k=0}^{K} \gamma_k \boldsymbol{L}^k \boldsymbol{H}$ with the $K + 1$ diffusion coefficients $\boldsymbol{\gamma} \in \mathbb{R}^{K+1}$. Many GNNs stack multiple convolutions and add point-wise non-linearities.

Crucially, GCN's or APPNP's graph filter $g(\boldsymbol{\Lambda})$ is fixed (up to scaling). Specifically, we obtain an MLP with $\boldsymbol{\gamma} = [1, 0, \ldots, 0]$. If using $\mathring{\boldsymbol{L}}$, a GCN corresponds to $\boldsymbol{\gamma} = [0, -1, 0, \ldots, 0]$, and in APPNP $\boldsymbol{\gamma}$ are the Personalized PageRank coefficients.

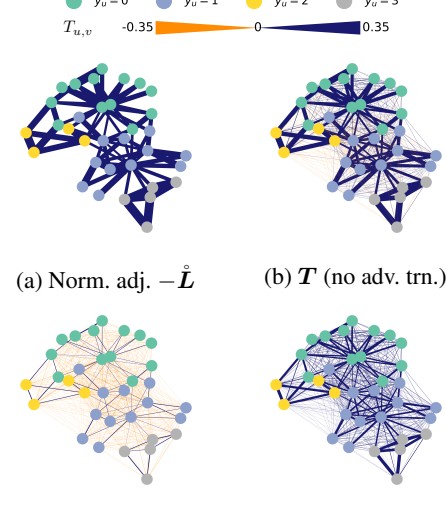

(a) Norm. adj. $-\mathring{\boldsymbol{L}}$      (b) $\boldsymbol{T}$ (no adv. trn.)

(c) $\boldsymbol{T}$ (w/o local con.)      (d) $\boldsymbol{T}$ (w/ local con.)

**Robust diffusion.** In contrast to prior work on robust graph learning, we do not solely use a static parametrization of $g(\boldsymbol{\Lambda})$. Instead, we learn the graph filter, which corresponds to training diffusion coefficients $\boldsymbol{\gamma}$. This allows the model to adapt the graph diffusion to the adversarial perturbations seen during training, i.e., we learn a *robust diffusion*.

For this, the used architectures consist of two steps: **(1)** using an MLP to preprocess the node features, i.e., $\boldsymbol{H} = f_\theta(\boldsymbol{X})$ where $\theta$ are learnable parameters; and then, **(2)** using a learnable diffusion computing the logits. For the learnable diffusion, we employ GPRGNN [13] that uses the previously introduced monomial basis: $\text{softmax}\left( \sum_{k=0}^{K} \gamma_k \mathring{\boldsymbol{L}}^k \boldsymbol{H} \right)$. Additionally, we study Chebyshev polynomials (see ChebNetII [14]) that are of interest due to their beneficial theoretical properties. ChebNetII can be expressed as $\text{softmax}\left( \sum_{k=0}^{K} \gamma_k w_k T_k(\boldsymbol{L}) \boldsymbol{H} \right)$ with extra weighting factor $w_k = {}^2/{}_{K-1} \sum_{j=0}^{K} T_k(x_j)$. The Chebyshev basis is given via $T_0(\boldsymbol{L}) = \boldsymbol{I}$, $T_1(\boldsymbol{L}) = \boldsymbol{L}$, and $T_k(\boldsymbol{L}) = 2\boldsymbol{L} T_{k-1}(\boldsymbol{L}) - T_{k-2}(\boldsymbol{L})$. The Chebyshev nodes $x_j = \cos\left( \frac{j+1/2}{K+1} \pi \right), j = 0, \ldots, K$ for $w_k$ reduce the so-called Runge phenomenon [14]. Note, the resulting Chebyshev polynomial can be expressed in monomial basis (up to $\boldsymbol{L}$ vs. $\mathring{\boldsymbol{L}}$) via expanding the $T_k(\boldsymbol{L})$ terms and collecting the powers $\boldsymbol{L}^k$.

Figure 4: Robust diffusion (GPRGNN) on Karate Club where the edge $(u, v)$ width encodes the diffusion coefficient $T_{u,v}$ learned during training. In (a) we show the normalized adjacency matrix. The other plots show the robust diffusion transition matrix: (b) w/o adversarial training, (c) w/ adversarial training but w/o local constraints, and (d) w/ adversarial training and w/ local constraints.

**Interpretability.** While chaining multiple layers of a "fixed" convolution scheme (e.g. GCN) might allow for similar flexibility, with our choice of robust diffusion, we can gain insights about the learned robust representation from the *(i)* polynomial, *(ii)* spectral, and *(iii)* spatial perspective.

*(i) Polynomial perspective.* The coefficients $\gamma_k$ determine the importance of the respective $k$-hop neighborhood for the learned representations. To visualize $\gamma_k$, we can always consider $\gamma_0$ to be positive, which is equivalent to flipping the sign of the processed features $\boldsymbol{H}$. Additionally, we normalize the coefficients s.t. $\sum |\gamma| = 1$ since $\boldsymbol{H}$ also influences the scale. In Figure 6, we give an example for the polynomial perspective (details are discussed in Section 5).

*(ii) Spectral perspective.* We solve for $g_\theta(\mathbf{\Lambda})$ in the polynomial approximation $\mathbf{V} g_\theta(\mathbf{\Lambda}) \mathbf{V}^\top \mathbf{H} \approx \sum_{k=0}^{K} \gamma_k \mathring{\mathbf{L}}^k \mathbf{H}$ to obtain a possible graph filter $g_\theta(\mathbf{\Lambda}) = \mathbf{V}^\top (\sum_{k=0}^{K} \gamma_k \mathring{\mathbf{L}}^k) \mathbf{V}$. Following Balcilar et al. [15], we study the spectral characteristics w.r.t. $\mathbf{I} - \mathbf{D}^{-1/2} \mathbf{A} \mathbf{D}^{-1/2}$. Then, the diagonal entries of $g_\theta(\mathbf{\Lambda})$ correspond to the (relative) magnitude of how signals of frequency $\lambda$ are present in the filtered signal. Vice versa, a low value corresponds to suppression of this frequency. Recall, low frequencies correspond to the small eigenvalues and high to the large eigenvalues. In Figure 7, we show how adversarial training and the permissible perturbations affect the spectra of the learned graph filters.

*(iii) Spatial perspective.* Robust diffusion (monomial basis) can be summarized as $\mathrm{softmax}\,(\mathbf{T}\mathbf{H})$ where $\mathbf{T} = \sum_{k=0}^{K} \gamma_k \mathring{\mathbf{L}}^k$ is the total diffusion matrix. The coefficient $\mathbf{T}_{uv}$ indicates the diffusion strength between node $u$ and node $v$. For example, we visualize the total diffusion matrix $\mathbf{T}$ in Figure 4 on Karate Club [16] with different training strategies. Depending on the learning signal we give, GPRGNN is able to adjust its diffusion.

## 4   LR-BCD: Adversarial Attack with Local Constraints

**Motivation for local constraints.** The just-discussed interpretability of robust diffusion provides empirical evidence for the importance of local constraints. Specifically, from Figure 4c, we see that GPRGNN adversarially trained *without* local constraints learns a diffusion that almost ignores the graph structure. While this model is certainly very robust w.r.t. structure perturbations, it is not a very useful GNN since it cannot leverage the structure information anymore. In contrast, we show in Figure 4d that GPRGNN trained adversarially *with* local constraints results in a robust model that can still incorporate the structure information.

The local predictions in node classification yield an alternative motivation. In the absence of local constraints, an adversary typically has the power to rewire the entire neighborhood of many nodes. When attacking GNNs, we empirically observe that a majority of successfully attacked nodes are perturbed beyond their degree even for moderate global attack budgets (see Figure 5). That such perturbations are not reasonable is evident in the fact that studies on local attacks [7, 17], where the adversary targets a single node's prediction, do not consider perturbations (far) beyond the node degree. However, a $5\%$ attack budget on Cora-ML allows changing 798 edges while the majority of nodes have a degree less than three.

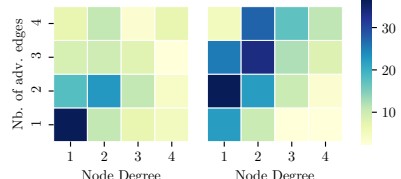

Figure 5: Number of successfully attacked nodes by node degree and number of connected adversarial edges. We attack self-trained GCN with PR-BCD and global budgets $10\%$ (left) and $25\%$ (right) on Cora-ML and aggregate the results over three different data splits. We observe that a notable amount of nodes is perturbed beyond their degree.

Traditionally, adversarial changes are judged by noticeability [18] since unnoticeable changes do not alter the semantics. However, for graphs, manual inspection of its content is often not feasible, and the concept of (un-)noticeability is unclear. However, using generative graph models Gosch et al. [6] revealed that perturbations beyond the node degree most often do alter the semantics. Perturbations that alter semantics can pose a problem for adversarial training. Similar observations have been done in graph contrastive learning where perturbations that do not preserve graph semantic preservation have a direct effect on the achievable error bounds [19].

**Locally constrained Randomized Block Coordinate Descent (LR-BCD).** We next introduce our LR-BCD attack; the first attack targeting multiple nodes at once while maintaining a local constraint for each node next to a global one. For this, we extend the so-called PR-BCD attack framework.

*Projected Randomized Block Coordinate Descent (PR-BCD)* [17] is a gradient-based attack framework applicable to a wide range of models. Its goals is to generate a perturbation matrix $\mathbf{P} \in \{0, 1\}^{n \times n}$ that is applied to the original adjacency matrix $\mathbf{A}$ to construct a perturbed matrix $\tilde{\mathbf{A}} = \mathbf{A} + (\mathbb{I} - 2\mathbf{A}) \odot \mathbf{P}$, where $\mathbb{I}$ is an all-one matrix and $\odot$ the element-wise product. For undirected graphs, only the upper-triangular parts of all matrices are considered. To construct $\mathbf{P}$ in an efficient manner, PR-BCD relaxes $\mathbf{P}$ during the attack from $\{0, 1\}^{n \times n}$ to $[0, 1]^{n \times n}$ and employs an iterative process for $T$ iterations. It consists of three repeating steps. **(1)** A random block of size $b$ is sampled. That is, only $b$ (non-contiguous) elements in the perturbation matrix $\mathbf{P}_{t-1}$ are considered and all other elements set to zero. Thus, $\mathbf{P}_{t-1}$ is sparse. **(2)** A gradient update w.r.t. the loss to compute relaxed perturbations

$S_t$ is performed, i.e., $S_t \leftarrow P_{t-1} + \alpha_{t-1} \nabla_{P_{t-1}} \ell(P_{t-1})$, where $P_{t-1}$ are the previous perturbations, $\alpha_{t-1}$ is the learning rate, and $\nabla_{P_{t-1}} \ell(P_{t-1})$ are the perturbation gradients through the GNN. Finally, and most crucially **(3)** a projection $\Pi_{\mathcal{B}(\mathcal{G})}$ ensures that the perturbations are *permissible* given the set of allowed perturbations $\mathcal{B}(\mathcal{G})$, i.e. $P_t = \Pi_{\mathcal{B}(\mathcal{G})}(S_t)$. Now, the process starts again with **(1)**, but all zero-elements in the block $b$ (or at least $1/2$ of the lowest-value block elements) are resampled. After $T$ iterations, the perturbations $P_T$ are discretized from $[0,1]^{n \times n}$ to $\{0,1\}^{n \times n}$ to obtain the discrete $\tilde{A}$. Specifically, $P_T$ is discretized via sampling, where the elements in $P_T$ are used to define Bernoulli distributions.

*Global projection.* As mentioned above, the projection $\Pi_{\mathcal{B}(\mathcal{G})}$ of PR-BCD is the crucial step that accounts for the attack budget. Geisler et al. [17] develop an efficient projection for an adversary implementing a global perturbation constraint, i.e., for $\mathcal{B}(\mathcal{G}) = \{\tilde{A} \in \{0,1\}^{n \times n} \mid \|\tilde{A} - A\|_0 \leq \Delta\}$ with $\Delta \in \mathbb{N}_{\geq 0}$. This is achieved by formulating the projection as the following optimization problem

$$\Pi_\Delta(S) = \arg\min_P \|P - S\|_F^2 \qquad \text{subject to} \quad \sum\nolimits_{i,j} P_{i,j} \leq \Delta \tag{5}$$
$$P \in [0,1]^{n \times n}$$

with Frobenius norm $\|\cdot\|_F^2$ and the sum aggregating all elements of the matrix. $\Pi_\Delta(S)$ can be solved with the bisection method [17]. However, the projection $\Pi_\Delta$ does not support limiting the number of perturbations per node, i.e., $\sum_{j=1}^n P_{i,j} \leq \Delta_i^{(l)}$ where $\mathbf{\Delta}^{(l)} \in \mathbb{N}_{\geq 0}^n$ is the vector of local budgets for each node. Indeed, extending the optimization problem above to include local constraints leads to the notoriously hard problem of Euclidean projection to polyhedra [20].

*A locally constrained global projection (LR-BCD).* With the goal of an efficient attack including local constraints, we have to develop an alternative and scaleable projection strategy for $\mathcal{B}(\mathcal{G}) = \{\tilde{A} \in \{0,1\}^{n \times n} \mid \|\tilde{A} - A\|_0 < \Delta \wedge \|\sum_j(\tilde{A}_{ij} - A_{ij})\|_0 \leq \Delta_i^{(l)} \forall i \in [n]\}$. Our novel projection $P = \Pi_\Delta^{(l)}(S) = \Pi_{[0,1]}(S) \odot C^*$ chooses the largest entries from the perturbation matrix $S \in \mathbb{R}^{n \times n}$ using $C^* \in [0,1]^{n \times n}$ and clipping operator $\Pi_{[0,1]}(s) \in [0,1]$, s.t. we obey global *and* local constraints. The optimal choices $C^*$ can be calculated by solving a relaxed multi-dimensional knapsack problem:

$$C^* = \arg\max_C \sum\nolimits_{i,j} S \odot C \qquad \text{subject to} \quad \sum\nolimits_{i,j} P'_{i,j} \leq \Delta \tag{6}$$
$$\sum\nolimits_j P'_{i,j} \leq \Delta_i^{(l)} \qquad \forall i \in [n]$$
$$C \in [0,1]^{n \times n}$$

where $P' = \Pi_{[0,1]}(S) \odot C$, i.e., $\Pi_{[0,1]}(S)$ represents the weights of the individual choices, while $S$ captures their value. $\sum_{i,j} P'_{i,j}$ aggregate all elements in the matrix. The local constraint $\sum_j P'_{i,j} \leq \Delta_i^{(l)}$, constrain the perturbations in each neighborhood, with node-specific budgets $\Delta_i^{(l)}$. Even though this optimization problem has no closed-form solution, it can be readily approximated with greedy approaches [21]. The key idea is to iterate the non-zero entries in $S$ in descending order and construct the resulting $C^*$ (or directly $P$) as follows: For each perturbation $(u,v)$ related to $S_{uv}$, we check if the global $\Delta$ or local budgets $\Delta_u^{(l)}$ and $\Delta_v^{(l)}$ are not yet exhausted. Then, $C_{uv}^* = 1$ (i.e., $P_{u,v} = \Pi_{[0,1]}(S_{uv})$). Otherwise, $C_{uv}^* = \min\{\Delta, \Delta_u^{(l)}, \Delta_v^{(l)}\}$. The final discretization is given by $C^*$ corresponding to the solution of Equation (6) for $S_T$, but changing the weight of each choice to 1 (i.e., $P' = C$), guaranteeing a binary solution due to the budgets being integer. We give the pseudo-code and more details in Appendix B.4.

Our projection yields sparse solutions as it greedily selects the largest entries in $S$ until the budget is exhausted. Moreover, it is efficient. Its time complexity is $\mathcal{O}(b \log(b))$ due to the sorting of the up to $b$ non-zero entries in the sparse matrix $S$. Its space complexity is $\mathcal{O}(b)$, assuming we choose the block size $b \geq \Delta$ as well as $b \geq n$. Thus, LR-BCD has the same asymptotic complexities as the purely global projection by Geisler et al. [17] (which we will from now on denote as PR-BCD).

**Summary.** We will now summarize our LR-BCD along the four dimensions proposed by Biggio and Roli [22]. *(1) Attacker's goal:* Although our focus is to increase the misclassification for the target nodes in node classification, we can apply LR-BCD to different node- and graph-level tasks depending on the loss choice. *(2) Attacker's knowledge:* We assume perfect knowledge about the model and dataset, which is reasonable for adversarial training. *(3) Attacker's capability:* We propose

Table 1: Comparison on Citeseer of regularly trained models, the state-of-the-art SoftMedian GDC defense, and adversarially trained models (train $\epsilon = 20\%$). The first line shows the robust accuracy [%] of a standard GCN and all other numbers represent the difference in robust accuracy achieved by various models in percentage points from this standard GCN. The best model is highlighted in bold and grey background marks our *robust diffusion*.

| Model | Adv. trn. | A. eval. → A. trn. ↓ | Clean | LR-BCD | PR-BCD | LR-BCD | PR-BCD | Certifiable accuracy / sparse smoothing | | |
| --- | --- | --- | --- | --- | --- | --- | --- | --- | --- | --- |
| | | | | $\epsilon=0.1$ | | $\epsilon=0.25$ | | Clean | 3 add. | 5 del. |
| GCN | ✗ | - | 72.0 ± 2.5 | 54.7 ± 2.8 | 51.7 ± 2.8 | 45.3 ± 3.4 | 38.0 ± 3.8 | 38.3 ± 11.5 | 1.7 ± 0.7 | 4.8 ± 1.5 |
| GAT | ✗ | - | -3.6 ± 2.7 | -3.9 ± 3.4 | +0.5 ± 3.5 | -15.9 ± 5.3 | -2.3 ± 7.3 | -14.0 ± 12.0 | -1.7 ± 0.7 | -4.8 ± 1.5 |
| APPNP | ✗ | - | +0.2 ± 1.1 | +1.7 ± 0.7 | +1.9 ± 1.4 | +3.0 ± 1.2 | +2.2 ± 2.5 | +8.9 ± 9.1 | +31.2 ± 6.4 | +31.6 ± 6.4 |
| GPRGNN | ✗ | - | +2.2 ± 4.3 | +4.2 ± 2.7 | +3.6 ± 4.9 | +5.5 ± 3.9 | +7.9 ± 4.6 | +17.9 ± 6.9 | +42.4 ± 4.4 | +41.3 ± 3.7 |
| ChebNetII | ✗ | - | +1.1 ± 2.2 | +5.8 ± 2.5 | +5.0 ± 2.4 | +10.4 ± 2.6 | +7.6 ± 3.4 | +24.6 ± 9.9 | +55.6 ± 1.2 | +54.0 ± 0.9 |
| SoftMedian | ✗ | - | +0.9 ± 1.7 | +9.5 ± 2.2 | +9.3 ± 1.9 | +16.2 ± 2.4 | +14.6 ± 2.9 | +25.2 ± 10.5 | +60.3 ± 1.4 | +57.5 ± 0.8 |
| GCN | ✓ | LR-BCD | -0.2 ± 1.2 | +7.8 ± 1.6 | +5.9 ± 1.5 | +10.9 ± 2.1 | +8.1 ± 2.3 | -3.0 ± 9.4 | +10.7 ± 4.6 | +13.1 ± 5.1 |
| | | PR-BCD | +0.0 ± 1.9 | +6.9 ± 0.9 | +5.3 ± 1.6 | +8.6 ± 2.3 | +5.8 ± 2.4 | +4.2 ± 14.6 | +10.1 ± 5.5 | +11.4 ± 4.5 |
| GAT | ✓ | LR-BCD | +0.8 ± 1.6 | +5.9 ± 3.7 | +9.0 ± 3.0 | +8.4 ± 5.1 | +13.1 ± 3.5 | -2.0 ± 23.0 | +1.4 ± 1.7 | +4.0 ± 2.1 |
| | | PR-BCD | +1.1 ± 2.2 | +8.9 ± 2.8 | +13.2 ± 3.7 | +10.3 ± 2.3 | +21.8 ± 4.8 | -10.1 ± 13.7 | +1.2 ± 2.0 | +2.8 ± 1.2 |
| APPNP | ✓ | LR-BCD | -0.8 ± 1.3 | +7.6 ± 1.6 | +5.6 ± 1.9 | +11.1 ± 1.9 | +8.3 ± 3.6 | +21.7 ± 9.5 | +41.3 ± 2.9 | +44.1 ± 1.2 |
| | | PR-BCD | -0.2 ± 2.2 | +6.2 ± 2.3 | +5.5 ± 3.1 | +9.0 ± 2.9 | +6.5 ± 3.8 | +19.3 ± 7.2 | +41.0 ± 3.8 | +41.9 ± 3.0 |
| GPRGNN | ✓ | LR-BCD | +1.7 ± 3.0 | **+15.7 ± 3.6** | +13.6 ± 3.5 | **+24.8 ± 4.2** | +22.9 ± 4.3 | **+34.0 ± 11.5** | +67.4 ± 1.5 | +65.0 ± 2.5 |
| | | PR-BCD | +0.6 ± 3.6 | +15.0 ± 3.6 | **+15.9 ± 4.2** | +23.2 ± 4.3 | **+26.3 ± 5.4** | +32.9 ± 11.8 | **+69.0 ± 1.5** | **+65.9 ± 2.3** |
| ChebNetII | ✓ | LR-BCD | **+3.7 ± 1.4** | +11.5 ± 1.6 | +11.5 ± 1.1 | +16.4 ± 1.9 | +16.0 ± 2.9 | +31.5 ± 12.7 | +62.3 ± 1.9 | +59.8 ± 2.5 |
| | | PR-BCD | +3.4 ± 1.7 | +13.2 ± 2.2 | +14.3 ± 1.2 | +19.5 ± 1.6 | +19.8 ± 2.3 | +30.7 ± 13.2 | +65.4 ± 2.6 | +62.9 ± 3.7 |

to use a threat model that constrains the number of edge perturbations globally as well as locally for each node. *(4) Attacker's strategy:* LR-BCD extends PR-BCD by a projection incorporating local constraints. Most notably, we relax the unweighted graph during the attack to continuous values and leverage randomization for an efficient attack while optimizing over all possible $n^2$ edges.

## 5 Empirical Results

**Inductive learning.** All results in this section follow a fully inductive semi-supervised setting. That is, the training graph $\mathcal{G}$ does not include validation and test nodes. For evaluation, we then successively add the respective evaluation set. This way, we avoid the evaluation flaw of prior work since the model cannot achieve robustness via "memorizing" the predictions on the clean graph. We obtain inductive splits randomly, except for OGB arXiv [11], which comes with a split. In addition to the commonly sampled 20 nodes per class for both (labeled) training and validation, we sample a stratified test set consisting of $10\%$ of all nodes. The remaining nodes are used as (unlabeled) training nodes.

**Setup.** We study the citation networks Cora, Cora-ML, CiteSeer [23], Pubmed [24], and OGB arXiv [11]. Furthermore, WikiCS [25, 26], which has significantly more heterogeneous neighborhoods as well the heterophilic dataset Squirrel [27] and (contextual) Stochastic Block Models (SBMs) [28] with a heterophilic parametrization (see Appendix B.1). As models we choose GPRGNN, ChebNetII, GCN, APPNP, and GAT [29]. Further, we compare to the state-of-the-art evasion defenses Soft Median GDC [17] in the main part and GRAND [30] in Table D.6. We apply adversarial training (see Section 2 and Appendix B.5) using both PR-BCD that only constraints perturbations globally and our *LR-BCD* that also allows for local constraints. Moreover, we use adversarial training in conjunction with self-training. Due to the inductive split, this does not bias results. We use the tanh margin attack loss of [17] and do not generate adversarial examples for the first 10 epochs (warm-up). We evaluate robust generalization on the test set using L/PR-BCD, which corresponds to *adaptive attacks*. Adaptive attacks are the gold standard in evaluating empirical robustness because they craft model-specific perturbations [31]. We use $\epsilon$ to parametrize the global budget $\Delta = \lfloor \epsilon \cdot \sum_{u \in \mathcal{A}} d_u / 2 \rfloor$ relative to the degree $d_u$ for the set of targeted nodes $\mathcal{A}$. We find that $\Delta_u^{(l)} = \lfloor d_u / 2 \rfloor$ is a reasonable local budget for all datasets but arXiv where we use $\Delta_u^{(l)} = \lfloor d_u / 4 \rfloor$. We report averaged results with the standard error of the mean over three random splits. We use GTX 1080Ti (11 Gb) GPUs for all experiments but arXiv, for which we use a V100 (40 GB). For details see Appendix B. We discuss limitations in Appendix F and provide code at `https://www.cs.cit.tum.de/daml/adversarial-training/`.

**Certifiable robustness.** We use the model-agnostic randomized/sparse smoothing certificate of Bojchevski et al. [32] to also quantify certifiable robustness. Sparse smoothing creates a randomized ensemble given a base model $f_\theta$ s.t. the majority prediction of the ensemble comes with guarantees. For the randomization, we follow Bojchevski et al. [32] and uniformly drop edges with $p_- = 0.6$ as well as add edges with $p_+ = 0.01$. Sparse smoothing then determines if a node-level prediction is

certified (does not change) for the desired deletion radius $r_-$ or addition radius $r_+$. The guarantee holds in a probabilistic sense with significance level $\alpha$. We report the "certified accuracy" $\gamma(r_-, r_+)$ that is the average of correct and certifiable predictions over all nodes. We choose $\alpha = 5\%$ and obtain 100,000 random samples. For simplicity, we report in the main part the certified accuracies $\gamma(r_- = 0, r_+ = 0)$, $\gamma(5, 0)$, and $\gamma(0, 3)$. See Appendix D.4 for more details and results.

**Finding I: Adversarial training is an effective defense against structure perturbations.** This is apparent from the results in Table 1, where we compare the empirical and certifiable robustness between the aforementioned models on Citeseer. Our adversarially trained robust diffusion models GPRGNN and Cheb-NetII outperform the other baselines both in empirical and certifiable robustness. This includes the state-of-the-art defense Soft Median, which we outperform with a comfortable margin. *Thus, we close the gap in terms of the efficacy of adversarial training in the image domain vs. structure perturbations.* Notably, the increased robustness does not imply a lower clean accuracy. For example, our LR-BCD adversarially trained GPRGNN achieves a 1.7 percentage points higher clean accuracy than a GCN, while with an adaptive LR-BCD attack and $\epsilon = 25\%$ perturbed edges, we outperform a GCN by 24.8 percentage points. This amounts to a clean accuracy of 73.7% and a perturbed accuracy of 70.1%. We show the empirical robustness gains of adversarial training on on the other datasets in Table 2, compared to a GCN and regularly trained GPRGNN. We show that *robust diffusion* consistently and substantially improves the robustness – not only on homophilic datasets but also under heterophily (SBM, Squirrel). We present more results and an ablation study in Appendix D.

Table 2: Robust accuracy [%] on further datasets. Attack types match for training ($\epsilon = 0.2$) and evaluation. Grey shading highlights our *robust diffusion*.

| | Model | Adv. trn. | Attack eval. & trn. | Clean | $\epsilon=0.1$ | $\epsilon=0.25$ |
|---|---|---|---|---|---|---|
| **Cora ML** | GCN | ✗ | LR-BCD | 82.5 ± 1.9 | 59.2 ± 2.8 | 48.5 ± 1.6 |
| | | | PR-BCD | 82.5 ± 1.9 | 57.4 ± 2.3 | 38.0 ± 2.3 |
| | GPRGNN | ✗ | LR-BCD | 83.5 ± 2.6 | 64.8 ± 2.3 | 57.0 ± 1.5 |
| | | | PR-BCD | 83.5 ± 2.6 | 61.9 ± 1.8 | 46.6 ± 1.4 |
| | | ✓ | LR-BCD | 83.3 ± 1.2 | 76.8 ± 1.1 | 74.4 ± 1.8 |
| | | | PR-BCD | 82.5 ± 0.8 | 73.5 ± 1.5 | 69.0 ± 2.6 |
| **Cora** | GCN | ✗ | LR-BCD | 79.9 ± 0.9 | 60.8 ± 0.2 | 47.9 ± 1.8 |
| | | | PR-BCD | 79.9 ± 0.9 | 59.1 ± 0.4 | 44.1 ± 0.7 |
| | GPRGNN | ✗ | LR-BCD | 81.8 ± 1.0 | 69.4 ± 0.2 | 61.7 ± 0.6 |
| | | | PR-BCD | 81.8 ± 1.0 | 64.3 ± 0.6 | 52.0 ± 0.6 |
| | | ✓ | LR-BCD | 82.4 ± 0.9 | 72.6 ± 1.1 | 70.0 ± 1.5 |
| | | | PR-BCD | 79.9 ± 1.3 | 71.9 ± 1.0 | 66.3 ± 0.9 |
| **Pubmed** | GCN | ✗ | LR-BCD | 77.4 ± 0.4 | 60.4 ± 0.7 | 52.8 ± 1.4 |
| | | | PR-BCD | 77.4 ± 0.3 | 54.0 ± 0.5 | 39.3 ± 1.2 |
| | GPRGNN | ✗ | LR-BCD | 78.8 ± 1.0 | 61.1 ± 1.3 | 54.0 ± 0.9 |
| | | | PR-BCD | 78.8 ± 1.0 | 57.0 ± 1.2 | 42.7 ± 1.4 |
| | | ✓ | LR-BCD | 80.5 ± 0.8 | 75.8 ± 1.0 | 74.7 ± 1.1 |
| | | | PR-BCD | 80.4 ± 0.5 | 73.0 ± 1.1 | 68.8 ± 1.6 |
| **SBM (hetero.)** | GCN | ✗ | LR-BCD | 62.0 ± 4.5 | 51.5 ± 4.3 | 45.1 ± 4.5 |
| | | | PR-BCD | 62.0 ± 4.5 | 49.2 ± 3.4 | 47.1 ± 4.8 |
| | GPRGNN | ✗ | LR-BCD | 86.5 ± 2.1 | 67.3 ± 3.3 | 59.6 ± 3.3 |
| | | | PR-BCD | 85.9 ± 4.9 | 68.4 ± 2.9 | 55.6 ± 4.4 |
| | | ✓ | LR-BCD | 85.5 ± 0.5 | 72.1 ± 1.7 | 67.0 ± 3.3 |
| | | | PR-BCD | 85.5 ± 2.9 | 69.0 ± 0.5 | 60.9 ± 2.9 |
| **Squirrel** | GCN | ✗ | LR-BCD | 42.0 ± 0.3 | 20.1 ± 1.3 | 15.8 ± 1.4 |
| | | | PR-BCD | 41.9 ± 0.3 | 12.1 ± 0.3 | 6.9 ± 0.6 |
| | GPRGNN | ✗ | LR-BCD | 40.0 ± 0.6 | 29.8 ± 4.8 | 28.4 ± 5.1 |
| | | | PR-BCD | 40.4 ± 0.9 | 20.6 ± 0.6 | 15.3 ± 1.4 |
| | | ✓ | LR-BCD | 37.8 ± 1.4 | 34.1 ± 3.4 | 33.8 ± 3.7 |
| | | | PR-BCD | 38.4 ± 1.6 | 32.3 ± 4.7 | 28.9 ± 9.1 |
| **WikiCS** | GCN | ✗ | LR-BCD | 74.6 ± 2.8 | 42.5 ± 2.3 | 35.4 ± 2.6 |
| | | | PR-BCD | 75.1 ± 1.5 | 38.6 ± 2.9 | 30.2 ± 2.5 |
| | GPRGNN | ✗ | LR-BCD | 72.8 ± 1.1 | 52.8 ± 3.2 | 49.7 ± 4.9 |
| | | | PR-BCD | 72.8 ± 0.4 | 52.7 ± 1.0 | 50.0 ± 1.2 |
| | | ✓ | LR-BCD | 73.3 ± 2.7 | 64.4 ± 2.2 | 62.8 ± 2.1 |
| | | | PR-BCD | 73.2 ± 0.2 | 60.0 ± 1.9 | 54.1 ± 0.8 |

**Finding II: Choose the set of permissible perturbations wisely!** As argued already in Section 4, the right set of admissible permutations can guide the learned robust diffusion. Importantly, the set of admissible perturbations is application specific. That is, depending on the choice, the resulting model might have different robustness characteristics w.r.t. different threat models. For example, in Table 1, a LR-BCD adversarially trained GPRGNN is more robust to LR-BCD attacks than to a model trained with PR-BCD, and vice versa. Importantly, the learned message passing is very different as the spectral analysis (Figure 7) reveals a striking pattern. Adversarial training with PR-BCD flipped the behavior from low-pass to high-pass if compared to a regularly trained model. However, adversarial training with LR-BCD seems to preserve the general characteristic while using a larger fraction of the spectrum to enhance robustness. Note that such a filter (less integral Lipschitz) opposes the theoretical stability result of Gama et al. [33]. Moreover, the polynomial coefficients (Figure 6) resulting from adversarial training without local constraints (Figure 6c) are very similar to coefficients that Chien et al. [13] associated with a non-informative graph structure. See Appendix D.2 for other datasets.

**Finding III: Adversarial training with local constraints is efficient and scalable.** We show in Figure 8 the results of an adversarially trained GPRGNN using LR-BCD with different relative budgets $\epsilon$. Adversarial training on arXiv requires less than 20 GB of RAM and an epoch takes approx. 10 seconds. In contrast, the globally constrained adversarial training of Xu et al. [2] would require multiple terabytes of RAM due to $n^2$ possible edge perturbations [17]. We not only show that robust diffusion with LR-BCD is scalable, but also that it improves the robustness on arXiv. That is, twice as many edge perturbations are required to push the accuracy below that of an MLP, if comparing $\epsilon = 10\%$ to standard training.

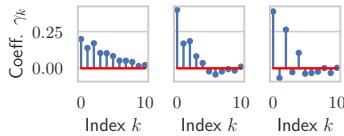

(a) Regul. (b) w/ loc. (c) w/o l.

Figure 6: Learned coefficients $\gamma$ for GPRGNN on Citeseer. We use $\epsilon = 20\%$ in (b) and (c) for adv. trn.

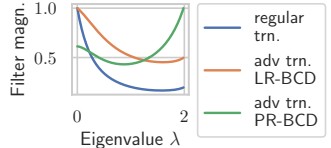

Figure 7: Respective spectral filters of Figure 6. Low filter magnitude denotes suppression of the respective frequencies associated with eigenvalues $\lambda$.

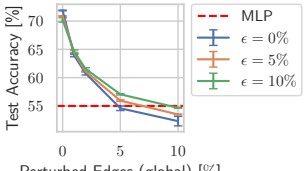

Figure 8: Acc. on arXiv under adv. attack for GPRGNN with different training pert. strengths $\epsilon$ using LR-BCD.

## 6 Related Work

We next relate to prior work and refer to Appendix E for more detail: *(1) Learning setting.* Conceptual problems by remembering training data are not unique to adversarial training and have been mentioned in Zheng et al. [40] for graph injection attacks and by Scholten et al.

Table 3: Works on adversarial training for GNNs.

| Publication | Learning setting | Type of attack |
|---|---|---|
| Deng et al. [34] | Transductive | evasion (attribute) |
| Feng et al. [35] | Transductive | evasion (attribute) |
| Jin and Zhang [36] | Transductive | evasion (structure + attribute) |
| Xu et al. [2] | Transductive | evasion (structure) |
| Xu et al. [3] | Transducitve | evasion (structure) |
| Chen et al. [37] | Transductive | evasion (structure) |
| Li et al. [38] | Transductive | evasion (structure + attribute) |
| Guo et al. [39] | Transductive | evasion + poisoning (structure) |

[41] for certification. *(2) Adversarial training* for GNNs under structure perturbations have been studied in [2, 3, 36–39]. However, all prior works study a transductive setting (see Table 3) having serious shortcomings for evasion attacks (see Section 2). Note that only Xu et al. [2, 3] study adversarial training for (global) structure perturbations in its literal sense, i.e., directly work on Equation (1). Further, we are the first to study adversarial training in the inductive setting. *(3) Local constraints* have been studied for local attacks [7, 17, 42, 12], i.e., if attacking a single node's prediction. They are also well-established for robustness certificates [43–45]. However, surprisingly, there is no global attack, i.e., attack targeting a large set of node predictions at once, that incorporates local constraints. This even led to prior work trying to naively and expensively apply local attacks for each node sequentially to generate locally constrained global perturbations [38]. With LR-BCD, we finally close this gap. *(4) GNN Architectures.* The robust GNN literature strongly focuses on GCN and variants [7, 17, 46–49, 38, 50, 8]. GAT is perhaps the most flexible studied model [40]. While adversarial training improves the robustness of GAT, in our experiments, an adversarial trained GAT is not more robust than a GCN [4]. A novel aspect of our work is that spectral filters, in the form of polynomials [14, 13], can learn significantly more robust representations, beating a state-of-the-art graph defense. They also reveal insights about the learned robust representation.

## 7 Broader impact

We are convinced that the benefits outweigh the risks. Having the right tools at hand can further the reliability of graph machine learning. Due to the more fine-grained perturbation models, researchers and practitioners and improve upon defending adversarial attacks. We firmly believe that transparent research into the vulnerabilities of models allows researchers and practitioners to understand potential problems and build strong defenses – as also showcased in our paper. Moreover, due to the studied whitebox setup, our approaches are not directly applicable for real-world malicious actors.

## 8 Discussion and Conclusion

We show that the transductive learning setting in prior work on adversarial training for GNNs has fundamental limitations leading to a biased evaluation and a trivial solution through memorization. Thus, we revisit adversarial training in a fully inductive setting. Furthermore, we argue that future research on evasion attacks in the graph domain, in general, should focus on the inductive setting to avoid the conceptual limitations inherent to combining transductive learning with test-time attacks. Moreover, we employ more flexible GNNs than before that achieve substantial robustness gains through adversarial training and are interpretable. For more realistic perturbations, we develop LR-BCD - the first global attack able to maintain local constraints.

## Acknowledgments and Disclosure of Funding

This research was supported by the Helmholtz Association under the joint research school "Munich School for Data Science - MUDS". Furthermore, this paper has been supported by the DAAD programme Konrad Zuse Schools of Excellence in Artificial Intelligence, sponsored by the German Federal Ministry of Education and Research, and the German Research Foundation, grant GU 1409/4-1.

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

# A Inductive Setting

Here, we formally introduce the inductive node-classification problem and the arising saddle-point problem relevant for adversarial training in the graph domain. The formal presentation follows the setting of a growing graph, i.e., given a training graph $\mathcal{G}$ of size $n$ with labels $y \in \mathcal{Y}^m$, $m \leq n$, we sample a new graph $\mathcal{G}'$ with $k$ additional nodes from the underlying data-generating distribution $\mathcal{D}$ conditioned on $\mathcal{G}$, which we denote $(\mathcal{G}', y') \sim \mathcal{D}(\mathcal{G}, y)$, where $y' \in \mathcal{Y}^{n+k}$ includes the (unknown) labels of the newly added nodes. The newly added nodes are collected in the set $\mathcal{I}$. However, as will be clear later, the given formulation is general and comprises different inductive learning settings described at the end of this section.

For inductive node classification, we are now interested in the expected $0/1$-loss on the newly added nodes, given as:

$$\mathbb{E}_{(\mathcal{G}', y') \sim \mathcal{D}(\mathcal{G}, y)} \sum_{i \in \mathcal{I}} \ell_{0/1}(f_\theta(\mathcal{G}'_i, y'_i)) \tag{A.1}$$

Now, assuming an adversary performing a test-time (evasion) attack, one can write the robust classification error as

$$\mathcal{L}_{adv}(f_\theta) = \mathbb{E}_{(\mathcal{G}', y') \sim \mathcal{D}(\mathcal{G}, y)} \max_{\tilde{\mathcal{G}}' \in \mathcal{B}(\mathcal{G}')} \sum_{i \in \mathcal{I}} \ell_{0/1}(f_\theta(\mathcal{G}'_i, y'_i)) \tag{A.2}$$

with $\mathcal{B}(\mathcal{G}')$ representing the set of possible perturbed graphs the adversary can choose from.

In adversarial training, the goal now is to solve the following saddle-point problem:

$$\min_\theta \mathcal{L}_{adv}(f_\theta) \tag{A.3}$$

Because of the expectations in the losses (compared to the transductive setting defined by Equation (3) and Equation (4)) neither an optimal solution to the saddle point problem defined by Equation (A.3) nor a perfectly robust model achieving clean test accuracy as in Proposition 1 can be achieved through memorizing information from the training graph. Thus, inductive node classification represents a learning setting without the pitfalls presented in Section 2.1.

The formulation comprises the setting of an evolving graph, which does not necessarily grow but changes the already existing nodes and graph structure over time, by including the indices of the $n$ nodes present at training in the set $\mathcal{K}$. Another inductive learning setting samples $\mathcal{G}'$ as a completely new graph independent of $\mathcal{G}$, then the conditional sampling process reduces to $(\mathcal{G}', y') \sim \mathcal{D}$.

# B Experimental Details

This section summarizes the experimental setup, including datasets, models, attacks, and adversarial training. The code can be found at `https://www.cs.cit.tum.de/daml/adversarial-training/`.

## B.1 Datsets

We perform experiments on the commonly used citation networks Cora, Cora-ML, Citeseer, Pubmed, and OGB arXiv; as well as WikiCS, Squirrel (with removed duplicates) and (C)SBMs. We always extract the largest connected component for all datasets.

For all datasets, except OGB arXiv and the heterophilic datasets, the data is split as follows. In the transductive setting, we sample 20 nodes per class for both training and validation set. The remaining nodes constitute the test set. Similarly, in the inductive setting, we sample 20 nodes per class for both (labeled) training and validation set. Additionally, we sample a stratified test set consisting of 10% of all nodes. The remaining nodes are used as additional (unlabeled) training nodes. For OGB arXiv, we use the provided temporal split. For Squirrel, we use the version without duplicated nodes and the split both provided by Platonov et al. [27].

We parametrize the (C)SBMs following Gosch et al. [6]. In a (C)SBM, the labels $y_i$ are sampled uniformly from $\{0, 1\}$. Then, the respective node features are sampled from a $d$-dimensional Gaußian distribution $\mathcal{N}((2y_i - 1)\mu, \sigma I)$ with $\mu = \frac{K\sigma}{2\sqrt{d}} \in \mathbb{R}^d$. We set $\sigma = 1$ and $K = 1.5$, resulting in a regime, where graph structure is important for node classification [6]. Furthermore, the dimensionality $d$ is set to 21. The connection probabilities between same-class nodes $p$ and different-class nodes $q$ are set to $p = 0.15\%$ and $q = 0.63\%$. (This can be understood as an "inverted" Cora fit, as the maximum likelihood fit to Cora results in $p = 0.63\%$ and $q = 0.15\%$.) We use an $80\%/20\%$ train/validation split on the nodes.

## B.2 Models

In the following, we present the hyperparameters and architectural details for the models used in this work. The experimental configuration files including all hyperparameters will be made public upon acceptance.

- GPRGNN: We fix the predictive part as a two-layer MLP with 64 hidden units (256 for OGB arXiv). The transition matrix is the symmetric normalized adjacency with self-loops $\mathring{L} = \mathring{D}^{-1/2}\mathring{A}\mathring{D}^{-1/2}$, while the diffusion coefficients are randomly initialized. A total of $K = 10$ diffusion steps is considered. During training, the MLP dropout is fixed at 0.2 and no dropout is applied to the adjacency. In contrast to Chien et al. [13], the diffusion coefficients are always learned with weight decay. This acts as regularization and prevents the coefficients from growing indefinitely.

- ChebNetII: Similarly, we use a two-layer MLP for the features with 64 hidden units and $K = 10$. Following the authors He et al. [14], we use the shifted graph Laplacian without self-loops $L = -D^{-1/2}AD^{-1/2}$. $L$ then has eigenvalues in the range $[-1, 1]$ instead of $[0, 2]$. This shift is required for the parametrization of the Chebyshev polynomial, however, does not affect eigenvectors. During training, we apply dropout of 0.8 to the features and 0.5 to $L$ to the propagation.

- GAT: We use two layers with 64 hidden units and a single attention head. During training we apply a dropout of 0.5 to the hidden units. No neighborhood dropout is applied.

- GCN: We use a two-layer GCN with 64 hidden units. For OGB arXiv we increase to a three layer architecture with 256 hidden units. During training, a dropout of 0.5 is applied.

- APPNP: The structure of APPNP mirrors that of GPRGNN. A two-layer MLP with 64 hidden units (256 for OGB arXiv) encodes the node attributes. This is followed by generalized graph diffusion with transition matrix $\mathring{L} = \mathring{D}^{-1/2}\mathring{A}\mathring{D}^{-1/2}$ and coefficient $\gamma_K = (1 - \alpha)K$ and $\gamma_l = \alpha(1 - \alpha)l$ for $l < K$. As for GPRGNN, a total of $K = 10$ steps is done and the MLP dropout is fixed at 0.2, while no dropout is applied to the adjacency. The $\alpha$ is set to 0.1.

- SoftMedian GDC: We follow the default configuration of Geisler et al. [17] and use a temperature of $T = 0.2$ for the SoftMedian aggregation with 64 hidden dimensions as well as dropout of 0.5. The Personalized PageRank (PPR) diffusion coefficient is fixed to $\alpha = 0.15$ and then a top $k = 64$ sparsification is applied. In the attacks, the model is fully differentiable but the sparsification of the propagation matrix.

- MLP: The MLP follows GPRGNN's prediction module. It has two layers with 64 hidden units (256 for OGB arXiv) and is trained using a dropout of 0.2 on the hidden layer.

**Training parameters.** We perform adversarial training as described in Algorithm B.2. We train for a maximum of 3000 epochs and optimize the model parameters using ADAM with learning rate 1e-2 and weight decay 1e-3. To avoid overfitting, we apply early stopping with a patience of 200 epochs. For early stopping, we attack all validation nodes with the training attack and consecutively evaluate the loss on all validation nodes w.r.t the resulting perturbed graph. Following insights from Geisler et al. [17], the tanh-margin loss is chosen as both attack and training loss. To increase stability, the first ten epochs are performed without including adversarial examples.

**Training attack parameters.** During adversarial training, we run a total of 20 attack epochs without early stopping. For PGD and PR-BCD, we stick with the same learning rates as for evaluation. For LR-BCD, we multiply the learning rate by 20 to account for the lower number of epochs. This helps to obtain a sparser solution where local constraints are accounted for.

## B.3 Attacks

For evaluating the robustness of GNNs, we use the following attacks and hyperparameters. Following insights from Geisler et al. [17], the tanh-margin loss is chosen as attack objective.

- PGD [3]: We optimize for 400 epochs. In epoch t, we use a learning rate of $0.1 \cdot \Delta/\sqrt{t}$, where $\Delta$ is the global budget (see Section 5).

- PR-BCD [17]: We closely follow the setup from Geisler et al. [17]. We use a block size of 500.000 (3.000.000 for OGB arXiv) and perform 400 epochs. Afterwards, the best epoch's state is recovered and 100 additional epochs with decaying learning rate and without block resampling are performed. Additionally, we scale the learning rate w.r.t. $\Delta$ and the block size, as suggested by Geisler et al. [17].

- LR-BCD: We use a block size of 500.000 (3.000.000 for OGB arXiv) and perform 400 epochs. We scale the learning rate w.r.t. $\Delta$ and the block size, identical to PR-BCD. The local budget is always chosen as

- FGSM [17]: The FGSM attack greedily perturbs a single edge each epoch. No hyperparameters are needed.

## B.4 Projection of LR-BCD

We next give the pseudo-code for our LR-BCD in Algorithm B.1. To improve performance, we only iterate edges that violate a local budget. Edges that only impact the global budget are handled separately. Additionally, we stop Algorithm B.1 early if the global budget is exhausted.

Note that the algorithm represents the implementation for undirected graphs.

---

**Algorithm B.1** LR-BCD - Projection with local and global constraints

---

**Input:** Upper triangular perturbations $\boldsymbol{S} \in \mathbb{R}^{n \times n}$, budgets $\Delta \in \mathbb{N}$, $\boldsymbol{\Delta}^{(l)} \in \mathbb{N}^n$
**Output:** projected gradients $\boldsymbol{P} = \Pi_\Delta^{(l)}(\boldsymbol{S})$

$\boldsymbol{P} \leftarrow \boldsymbol{0}$          // sparse matrix of shape $n \times n$
$\hat{\boldsymbol{S}} \leftarrow \Pi_{[0,1]}(\boldsymbol{S})$
idx $\leftarrow$ `argsort(triu(`$\boldsymbol{S}$`), 'desc', 'indices')`
**for** $(u, v)$ in idx **do**
    **if** $\hat{S}_{u,v} = 0$ or $\Delta - \hat{S}_{u,v} < 0$ **then** Return **end if**
    $\Delta_{u,v} \leftarrow \min(\Delta, \Delta_u^{(l)}, \Delta_v^{(l)})$
    **if** $\Delta_{u,v} \geq \hat{S}_{u,v}$ **then**
        $P_{u,v} \leftarrow \hat{S}_{u,v}$
        $\Delta \leftarrow \Delta - \hat{S}_{u,v}$
        $\Delta_u^{(l)} \leftarrow \Delta_u^{(l)} - \hat{S}_{u,v}$
        $\Delta_v^{(l)} \leftarrow \Delta_v^{(l)} - \hat{S}_{u,v}$
    **end if**
**end for**

---

## B.5 Adversarial training

In adversarial training, we solve the following objective

$$\arg\min_\theta \max_{\tilde{\mathcal{G}} \in B(\mathcal{G})} \sum_{i=1}^m \ell(f_\theta(\tilde{\mathcal{G}})_i, y_i) \tag{B.1}$$

where $m$ is the number of labeled nodes and $\mathcal{G}$ the clean training graph and $\ell$ is the chosen (surrogate) loss. Since Equation (1) is a challenging (typically) non-convex bi-level optimization problem, we approximate it with alternating first-order optimization. In other words, we train $f_\theta$ not on the clean graph $\mathcal{G}$ but on adversarially perturbed graphs $\tilde{\mathcal{G}}$ that are crafted in each iteration. This process is represented by Algorithm B.2.

---

**Algorithm B.2** Adversarial Training

---

**Input:** Training/validation graph $\mathcal{G}_{t/v}$, training/validation labels $\boldsymbol{y}_{t/v}$, GNN $f_{\theta_0}$, adversary $\mathcal{A}$, epochs $E$, warm-up epochs $W$, loss $\ell$, learning rate $\alpha$
**Output:** GNN $f_{\theta^*}$
$\ell_{min} \leftarrow \infty$
**for** $l = 1$ **to** $W$ **do**
   $\theta_l \leftarrow \theta_{l-1} + \alpha * \nabla_{\theta_{l-1}} \ell(f_{\theta_{l-1}}(\mathcal{G}_t), \boldsymbol{y}_t)$
**end for**
**for** $l = W$ **to** $E$ **do**
   $\hat{\mathcal{G}}_t \leftarrow \mathcal{A}(f_{\theta_{l-1}}, (\mathcal{G}_t), \boldsymbol{y}_t)$
   $\theta_l \leftarrow \theta_{l-1} + \alpha * \nabla_{\theta_{l-1}} \ell(f_{\theta_{l-1}}(\hat{\mathcal{G}}_t), \boldsymbol{y}_t)$
   $\tilde{\mathcal{G}}_v \leftarrow \mathcal{A}(f_{\theta_l}, \mathcal{G}_v, \boldsymbol{y}_v)$
   **if** $\ell_{min} > \ell(f_{\theta_l}(\hat{\mathcal{G}}_v), \boldsymbol{y}_v)$ **then**
      $\ell_{min} \leftarrow \ell(f_{\theta_l}(\hat{\mathcal{G}}_v), \boldsymbol{y}_v)$
      $\theta^* \leftarrow \theta_l$
   **end if**
**end for**

---

## B.6 Self-Training

For self-training, first pseudo-labels are generated using Algorithm B.3. Then, a new, final GNN is trained on the expanded label set.

---

**Algorithm B.3** Self-Training

---

**Input:** Graph $\mathcal{G}$, labels $\boldsymbol{y}^L$, GNN $f_\theta$, epochs $E$
**Output:** Expanded labels $\hat{\boldsymbol{y}}$
$f_{\theta^*} \leftarrow \texttt{train}(f_\theta, \mathcal{G}, \boldsymbol{y}^L)$           // train an initial GNN
$\boldsymbol{y}^U \leftarrow f_{\theta^*}(\mathcal{G})$           // predict pseudo labels for all unlabeled nodes
$\hat{\boldsymbol{y}} \leftarrow \boldsymbol{y}^L \cup \boldsymbol{y}^U$           // return union

---

# C  Transductive Setting

## C.1  Proof of Proposition 2

Before proving Proposition 2, note that the assumption $\mathcal{G} \in \mathcal{B}(\mathcal{G})$ in Proposition 2 is natural and usually fulfilled in practice. This is as the idea behind choosing a perturbation model $\mathcal{B}(\mathcal{G})$ is to include all graphs, which are close under some notion to the clean graph $\mathcal{G}$. As a result, all common perturbation model choices [12] allow the adversary to not change the graph if it would not increase misclassification, i.e., $\mathcal{G} \in \mathcal{B}(\mathcal{G})$ holds. Now, we restate Proposition 2 for convenience:

**Proposition 2.** *Assuming we solve the standard learning problem* $\theta^* = \arg\min_\theta \mathcal{L}_{0/1}(f_\theta)$ *and that* $\mathcal{G} \in \mathcal{B}(\mathcal{G})$. *Then,* $\tilde{f}_{\theta^*} = \mathcal{A}(f_{\theta^*})$ *is an optimal solution to the saddle-point problem arising in (transductive) adversarial training, in the sense of* $\mathcal{L}_{adv}(\tilde{f}_{\theta^*}) = \min_\theta \mathcal{L}_{adv}(\tilde{f}_\theta) \leq \min_\theta \mathcal{L}_{adv}(f_\theta)$.

*Proof.* Assuming $\mathcal{G} \in \mathcal{B}(\mathcal{G})$ leads to the following lower bound on the adversarial misclassification rate: $\mathcal{L}_{0/1}(f_\theta) \leq \mathcal{L}_{adv}(f_\theta)$, valid for all $\theta$. This implies $(i)$ $\min_\theta \mathcal{L}_{0/1}(f_\theta) \leq \min_\theta \mathcal{L}_{adv}(f_\theta)$. Now, assuming $\theta^* = \arg\min_\theta \mathcal{L}_{0/1}(f_\theta)$ is given and using our modified learning algorithm $\tilde{f}_{\theta^*}$, we obtain

$$\mathcal{L}_{adv}(\tilde{f}_{\theta^*}) = \min_\theta \mathcal{L}_{adv}(\tilde{f}_\theta) \leq \min_\theta \mathcal{L}_{adv}(f_\theta)$$

The first equality follows from $\mathcal{L}_{adv}(\tilde{f}_{\theta^*}) = \mathcal{L}_{0/1}(\tilde{f}_{\theta^*}) = \mathcal{L}_{0/1}(f_{\theta^*}) = \min_\theta \mathcal{L}_{0/1}(f_\theta)$ and then, using $\mathcal{L}_{0/1}(f_\theta) = \mathcal{L}_{0/1}(\tilde{f}_\theta) = \mathcal{L}_{adv}(\tilde{f}_\theta)$ results in $\min_\theta \mathcal{L}_{0/1}(f_\theta) = \min_\theta \mathcal{L}_{adv}(\tilde{f}_\theta)$. The last inequality follows from $(i)$. This shows the optimality of $\tilde{f}_{\theta^*}$ w.r.t. the saddle-point problem. 

$\square$

## C.2 Empirical Limitations

### C.2.1 Adversarial Training Causes Overfitting

Table C.1 shows that GPRGNN adversarially trained (robust diffusion) with $\epsilon = 1$ on Cora (with self-training) is (almost) perfectly robust against a wide range of attacks, not only PGD. Similar results hold for other common transductive benchmark datasets such as Cora-ML (Table C.2), or Citeseer (Table C.3).

Table C.1: GPRGNN's test-accuracy [%] under attack on Cora. This robust diffusion has been adversarially trained with $\epsilon = 1$ (with self-training) and shows (almost) perfect robustness against a wide range of attacks.

| Perturbed Edges [%] | 0 | 1 | 5 | 1 | 25 | 50 | 100 |
|---|---|---|---|---|---|---|---|
| PGD | 82.0+1.5 | 81.8+1.4 | 81.8+1.4 | 81.8+1.4 | 81.8+1.4 | 81.7+1.4 | 81.7+1.3 |
| FGSM | 82.0+1.5 | 81.8+1.4 | 81.8+1.4 | 81.7+1.3 | 81.3+1.2 | 81.0+1.1 | 80.6+1.0 |
| GreedyRBCD | 82.0+1.5 | 81.8+1.4 | 81.8+1.4 | 81.7+1.3 | 81.4+1.2 | 81.0+1.1 | 80.7+1.1 |
| PRBCD | 82.0+1.5 | 81.8+1.4 | 81.8+1.4 | 81.7+1.4 | 81.6+1.3 | 81.5+1.3 | 81.4+1.2 |

Table C.2: GPRGNN's test-accuracy [%] under attack on Cora-ML. This robust diffusion has been adversarially trained with $\epsilon = 1$ (with self-training) and shows (almost) perfect robustness against a wide range of attacks.

| Perturbed Edges [%] | 0 | 1 | 5 | 1 | 25 | 50 | 100 |
|---|---|---|---|---|---|---|---|
| PGD | 81.8+1.9 | 81.8+1.9 | 81.8+1.9 | 81.8+1.9 | 81.8+1.9 | 81.7+1.9 | 81.7+1.9 |
| DICE | 81.8+1.9 | 81.8+1.9 | 81.8+1.9 | 81.8+1.9 | 81.8+1.9 | 81.8+1.9 | 81.8+1.9 |
| FGSM | 81.8+1.9 | 81.8+2.0 | 81.7+1.9 | 81.6+2.0 | 81.5+2.0 | 81.2+2.0 | 80.9+2.0 |
| GreedyRBCD | 81.8+1.9 | 81.8+1.9 | 81.7+1.9 | 81.6+1.9 | 81.5+1.9 | 81.2+2.0 | 81.0+2.0 |
| PRBCD | 81.8+1.9 | 81.8+1.9 | 81.8+1.9 | 81.8+1.9 | 81.7+1.9 | 81.6+1.9 | 81.5+1.9 |

Table C.3: GPRGNN's test-accuracy [%] under attack on Citeseer. This robust diffusion has been adversarially trained with $\epsilon = 1$ (with self-training) and shows (almost) perfect robustness against a wide range of attacks.

| Perturbed Edges [%] | 0 | 1 | 5 | 1 | 25 | 50 | 100 |
|---|---|---|---|---|---|---|---|
| PGD | 71.8+0.9 | 71.8+0.9 | 71.8+0.9 | 71.8+0.9 | 71.7+0.9 | 71.8+0.8 | 71.8+0.9 |
| DICE | 71.8+0.9 | 71.8+0.9 | 71.8+0.9 | 71.8+0.9 | 71.8+0.9 | 71.8+0.9 | 71.9+0.9 |
| FGSM | 71.8+0.9 | 71.8+0.9 | 71.7+0.9 | 71.7+0.9 | 71.5+0.9 | 71.3+1.0 | 71.0+1.0 |
| GreedyRBCD | 71.8+0.9 | 71.8+0.9 | 71.7+0.9 | 71.7+0.9 | 71.5+0.9 | 71.3+0.9 | 71.0+1.0 |
| PRBCD | 71.8+0.9 | 71.8+0.9 | 71.8+0.9 | 71.7+0.9 | 71.7+0.9 | 71.7+0.9 | 71.4+0.9 |

### C.2.2 Self-Training as the (Main) Cause for Robustness

Figure C.1 shows that self-training is the major cause for robustness not only for GCN but also for other architectures. Figure C.2 highlights that this phenomenon also occurs given different attack strengths in the adversarial training. Figure C.3 shows that these results are not only constrained to the Cora dataset.

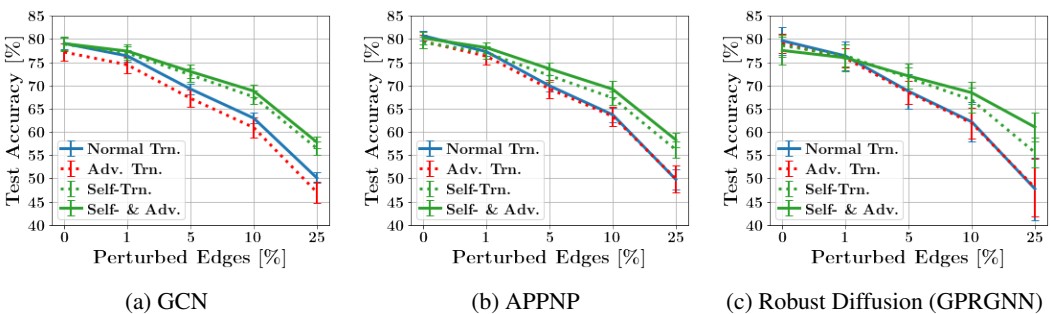

Figure C.1: Training different architectures with different training schemes on (transductive) Cora with a global training budget of $\epsilon = 0.1$ (10% perturbed edges) and using self-training.

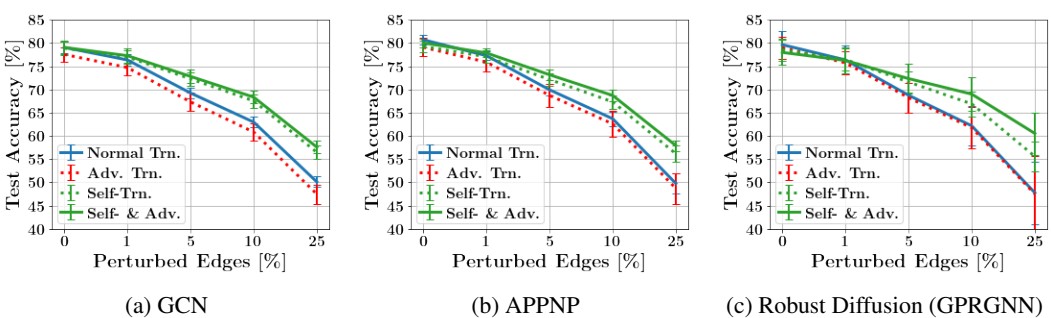

Figure C.2: Training different architectures with different training schemes on (transductive) Cora with a global training budget of $\epsilon = 0.05$ (5% perturbed edges) and using self-training.

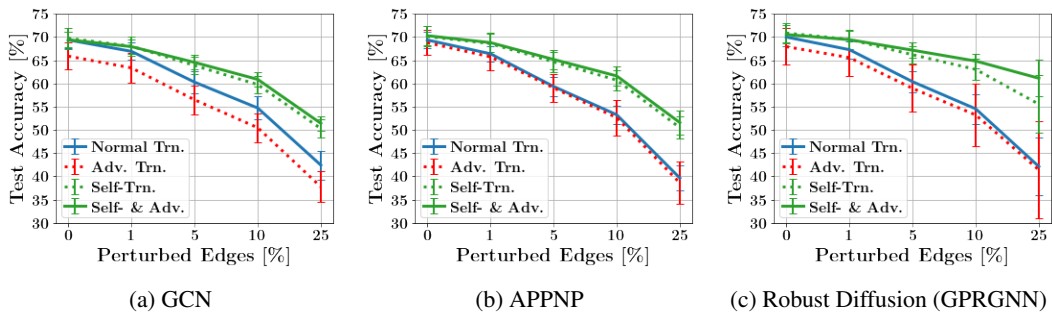

Figure C.3: Training different architectures with different training schemes on (transductive) Citeseer with a global training budget of $\epsilon = 0.1$ (10% perturbed edges) and using self-training.

# D   Additional Inductive Results

Here, we present additional results for adversarial training in the inductive setting. We supplement the results of the main part with results on other datasets as well as more extensive results for Citeseer.

## D.1   Test Accuracy

Table D.1 supplements Table 1 and additionally shows the results of the investigated models for an adversary with $\epsilon = 5\%$ and using only self-training without adversarial training. Thereby, we ablate the effects of self-training and adversarial training on the achieved robustness and highlight, that self-training alone already significantly increases robustness, but combining both methods achieves the best results.

Table D.2 and Table D.3 show analogous results for Cora and Cora-ML, respectively and Table D.4 for Pubmed. They again show that robust diffusion achieves the highest robustness values. Table D.5

Table D.1: Comparison on Citeseer of regularly trained models, the state-of-the-art SoftMedian GDC defense, self-trained models and adversarially trained models (train $\epsilon = 20\%$). The first line shows the robust accuracy [%] of a standard GCN and all other numbers represent the difference in robust accuracy achieved by various models in percentage points from this standard GCN. The best model is highlighted in bold.

| Model | Self-trn. | Adv. trn. | A. eval. → A. trn. ↓ | Clean | LR-BCD $\epsilon$=5% | PR-BCD | LR-BCD $\epsilon$=10% | PR-BCD | LR-BCD $\epsilon$=25% | PR-BCD |
|---|---|---|---|---|---|---|---|---|---|---|
| GCN | ✗ | ✗ | - | 72.0 ± 2.5 | 62.5 ± 1.9 | 59.3 ± 2.3 | 54.7 ± 2.8 | 51.7 ± 2.8 | 45.3 ± 3.4 | 38.0 ± 3.8 |
| GAT | ✗ | ✗ | - | -3.6 ± 2.7 | -3.6 ± 2.4 | -0.6 ± 3.0 | -3.9 ± 3.4 | +0.5 ± 3.5 | -15.9 ± 5.3 | -2.3 ± 7.3 |
| APPNP | ✗ | ✗ | - | +0.2 ± 1.1 | -0.9 ± 0.7 | +1.6 ± 0.7 | +1.7 ± 0.7 | +1.9 ± 1.4 | +3.0 ± 1.2 | +2.2 ± 2.5 |
| GPRGNN | ✗ | ✗ | - | +2.2 ± 4.3 | +2.0 ± 3.3 | +3.1 ± 4.4 | +4.2 ± 2.7 | +3.6 ± 4.9 | +5.5 ± 3.9 | +7.9 ± 4.6 |
| ChebNetII | ✗ | ✗ | - | +1.1 ± 2.2 | +4.0 ± 1.0 | +3.4 ± 2.5 | +5.8 ± 2.5 | +5.0 ± 2.4 | +10.4 ± 2.6 | +7.6 ± 3.4 |
| SoftMedian | ✗ | ✗ | - | +0.9 ± 1.7 | +4.4 ± 1.5 | +5.9 ± 1.4 | +9.5 ± 2.2 | +9.3 ± 1.9 | +16.2 ± 2.4 | +14.6 ± 2.9 |
| GCN | ✓ | ✗ | - | +1.1 ± 0.9 | +3.0 ± 0.3 | +3.4 ± 0.7 | +6.4 ± 0.9 | +5.1 ± 0.8 | +9.2 ± 1.4 | +7.5 ± 1.6 |
| GAT | ✓ | ✗ | - | +0.2 ± 2.6 | +0.8 ± 2.5 | +4.8 ± 2.7 | +1.6 ± 3.2 | +5.5 ± 3.7 | -6.7 ± 3.6 | +10.0 ± 4.3 |
| APPNP | ✓ | ✗ | - | -0.5 ± 2.2 | +2.0 ± 0.9 | +3.3 ± 1.3 | +6.2 ± 2.2 | +5.3 ± 2.3 | +8.7 ± 2.5 | +5.9 ± 3.3 |
| GPRGNN | ✓ | ✗ | - | +1.7 ± 3.4 | +7.3 ± 2.8 | +7.2 ± 3.4 | +13.2 ± 3.5 | +10.9 ± 3.7 | +18.4 ± 4.2 | +19.6 ± 5.4 |
| ChebNetII | ✓ | ✗ | - | +0.2 ± 1.7 | +2.6 ± 0.9 | +3.9 ± 1.5 | +7.6 ± 2.2 | +6.9 ± 1.3 | +12.0 ± 1.1 | +10.4 ± 2.3 |
| GCN | ✓ | ✓ | LR-BCD | -0.2 ± 1.2 | +2.3 ± 0.3 | +3.0 ± 0.9 | +7.8 ± 1.6 | +5.9 ± 1.5 | +10.9 ± 2.1 | +8.1 ± 2.3 |
| | | | PR-BCD | +0.0 ± 1.9 | +1.4 ± 0.3 | +2.5 ± 0.9 | +6.9 ± 0.9 | +5.3 ± 1.6 | +8.6 ± 2.3 | +5.8 ± 2.4 |
| GAT | ✓ | ✓ | LR-BCD | +0.8 ± 1.6 | +1.7 ± 2.0 | +5.1 ± 2.7 | +5.9 ± 3.7 | +9.0 ± 3.0 | +8.4 ± 5.1 | +13.1 ± 3.5 |
| | | | PR-BCD | +1.1 ± 2.2 | +4.5 ± 1.8 | +7.8 ± 1.2 | +8.9 ± 2.8 | +13.2 ± 3.7 | +10.3 ± 2.3 | +21.8 ± 4.8 |
| APPNP | ✓ | ✓ | LR-BCD | -0.8 ± 1.3 | +2.2 ± 1.0 | +3.0 ± 1.5 | +7.6 ± 1.6 | +5.6 ± 1.9 | +11.1 ± 1.9 | +8.3 ± 3.6 |
| | | | PR-BCD | -0.2 ± 2.2 | +2.2 ± 1.2 | +3.1 ± 1.8 | +6.2 ± 2.3 | +5.5 ± 3.1 | +9.0 ± 2.9 | +6.5 ± 3.8 |
| GPRGNN | ✓ | ✓ | LR-BCD | +1.7 ± 3.0 | **+8.1 ± 2.7** | +8.4 ± 3.4 | **+15.7 ± 3.6** | +13.6 ± 3.5 | **+24.8 ± 4.2** | +22.9 ± 4.3 |
| | | | PR-BCD | +0.6 ± 3.6 | +8.1 ± 2.8 | **+10.3 ± 3.4** | +15.0 ± 3.6 | **+15.9 ± 4.2** | +23.2 ± 4.3 | **+26.3 ± 5.4** |
| ChebNetII | ✓ | ✓ | LR-BCD | **+3.7 ± 1.4** | +6.7 ± 0.8 | +8.4 ± 1.0 | +11.5 ± 1.6 | +11.5 ± 1.1 | +16.4 ± 1.9 | +16.0 ± 2.9 |
| | | | PR-BCD | +3.4 ± 1.7 | +7.0 ± 0.9 | +10.0 ± 1.3 | +13.2 ± 2.2 | +14.3 ± 1.2 | +19.5 ± 1.6 | +19.8 ± 2.3 |

shows the results for OGB-arXiv. The training split in OGB-arXiv is fully labeled and hence, self-training is not applicable. Adversarial training improves the robustness of the investigated models on most occasions. A learnable, robust diffusion is particularly effective for small $\epsilon$, while fixating the diffusion can be advantageous for large $\epsilon$. This is most likely explained by the different structure of this large-scale graph compared to the other benchmark graphs, indicated by adversaries with $\epsilon = 1\%$ or $\epsilon = 2\%$ being already very effective and corresponding to a large number of adversarial edges.

In Table D.6, we additionally compare the performance using the DICE and PGD attacks. Moreover, we show that our adversarial training approach is a more effective defense than GRAND [30] against LR-BCD.

Table D.2: Comparison on Cora of regularly trained models, the state-of-the-art SoftMedian GDC defense, self-trained models and adversarially trained models (train $\epsilon = 20\%$). The first line shows the robust accuracy [%] of a standard GCN and all other numbers represent the difference in robust accuracy achieved by various models in percentage points from this standard GCN. The best model is highlighted in bold.

| Model | Self-trn. | Adv. trn. | A. eval. → A. trn. ↓ | Clean | LR-BCD $\epsilon$=5% | PR-BCD | LR-BCD $\epsilon$=10% | PR-BCD | LR-BCD $\epsilon$=25% | PR-BCD |
|---|---|---|---|---|---|---|---|---|---|---|
| GCN | ✗ | ✗ | - | 79.9 ± 0.9 | 67.5 ± 0.4 | 66.7 ± 0.8 | 60.8 ± 0.2 | 59.1 ± 0.4 | 47.9 ± 1.8 | 44.1 ± 0.7 |
| GAT | ✗ | ✗ | - | -2.7 ± 1.6 | -2.1 ± 0.9 | -1.8 ± 2.3 | -5.3 ± 1.0 | -1.1 ± 3.2 | -11.7 ± 3.9 | -0.6 ± 3.1 |
| APPNP | ✗ | ✗ | - | +1.6 ± 0.1 | +2.4 ± 1.7 | +2.2 ± 0.4 | +2.4 ± 1.7 | +2.7 ± 0.9 | +7.2 ± 2.6 | +3.1 ± 0.7 |
| GPRGNN | ✗ | ✗ | - | +2.0 ± 1.2 | +7.4 ± 0.5 | +4.0 ± 0.8 | +8.5 ± 0.1 | +5.3 ± 1.0 | +13.8 ± 1.4 | +7.9 ± 1.2 |
| ChebNetII | ✗ | ✗ | - | +1.0 ± 2.3 | +3.3 ± 2.1 | +1.1 ± 2.6 | +4.0 ± 1.3 | +2.1 ± 1.4 | +8.2 ± 0.7 | +4.9 ± 1.4 |
| SoftMedian | ✗ | ✗ | - | -2.9 ± 1.0 | +3.8 ± 1.8 | +2.2 ± 1.0 | +6.6 ± 1.1 | +5.9 ± 1.4 | +14.8 ± 2.7 | +10.4 ± 1.4 |
| GCN | ✓ | ✗ | - | +0.6 ± 1.0 | +3.5 ± 1.0 | +2.0 ± 0.9 | +5.4 ± 1.2 | +3.2 ± 0.6 | +9.4 ± 0.5 | +4.6 ± 0.8 |
| GAT | ✓ | ✗ | - | -4.2 ± 2.7 | -0.5 ± 1.7 | -2.2 ± 3.2 | -3.7 ± 1.0 | -1.7 ± 3.3 | -8.8 ± 3.6 | -2.4 ± 3.4 |
| APPNP | ✓ | ✗ | - | +2.0 ± 0.9 | +5.3 ± 1.7 | +4.0 ± 0.8 | +8.2 ± 1.4 | +6.1 ± 1.0 | +13.4 ± 2.6 | +7.1 ± 0.8 |
| GPRGNN | ✓ | ✗ | - | **+2.8 ± 1.0** | **+8.2 ± 0.5** | +6.7 ± 0.2 | +12.0 ± 0.7 | +10.1 ± 1.3 | +19.7 ± 2.4 | +16.7 ± 3.0 |
| ChebNetII | ✓ | ✗ | - | +2.3 ± 0.5 | +5.7 ± 1.2 | +3.3 ± 0.8 | +6.8 ± 1.2 | +4.6 ± 0.9 | +11.7 ± 2.3 | +4.6 ± 1.1 |
| GCN | ✓ | ✓ | LR-BCD | +0.5 ± 0.7 | +3.3 ± 1.1 | +2.1 ± 0.4 | +5.7 ± 0.6 | +3.1 ± 0.3 | +10.4 ± 2.4 | +5.4 ± 1.2 |
| | | | PR-BCD | +1.1 ± 1.1 | +3.1 ± 0.6 | +2.3 ± 0.2 | +6.2 ± 0.8 | +3.8 ± 0.3 | +8.7 ± 1.8 | +5.0 ± 0.7 |
| GAT | ✓ | ✓ | LR-BCD | -1.5 ± 0.9 | +0.4 ± 1.3 | +0.6 ± 1.0 | +2.4 ± 1.6 | +2.3 ± 0.5 | +6.5 ± 3.0 | +7.3 ± 1.4 |
| | | | PR-BCD | -2.1 ± 0.9 | +3.8 ± 0.7 | +5.0 ± 1.9 | +5.1 ± 1.5 | +7.6 ± 1.9 | +5.9 ± 3.5 | +14.5 ± 3.0 |
| APPNP | ✓ | ✓ | LR-BCD | +1.8 ± 1.1 | +5.0 ± 1.2 | +3.9 ± 0.7 | +8.5 ± 0.4 | +6.8 ± 0.8 | +14.7 ± 2.7 | +8.2 ± 1.2 |
| | | | PR-BCD | +1.0 ± 0.7 | +3.7 ± 1.1 | +3.7 ± 0.6 | +6.6 ± 1.3 | +5.0 ± 1.1 | +12.7 ± 3.5 | +7.3 ± 0.2 |
| GPRGNN | ✓ | ✓ | LR-BCD | +2.6 ± 0.2 | +8.2 ± 0.6 | **+8.1 ± 0.4** | +11.8 ± 1.2 | +11.0 ± 0.0 | +22.1 ± 2.3 | +16.8 ± 1.5 |
| | | | PR-BCD | +0.0 ± 0.6 | +8.1 ± 1.1 | +8.1 ± 0.4 | **+13.3 ± 1.3** | **+12.8 ± 0.7** | **+24.2 ± 2.4** | **+22.2 ± 1.4** |
| ChebNetII | ✓ | ✓ | LR-BCD | -6.7 ± 9.6 | -1.2 ± 7.0 | -1.2 ± 6.7 | +1.2 ± 6.2 | +1.8 ± 4.8 | +10.0 ± 5.8 | +6.5 ± 1.4 |
| | | | PR-BCD | +1.5 ± 1.5 | +3.7 ± 1.8 | +4.8 ± 0.4 | +7.7 ± 1.5 | +6.7 ± 1.4 | +11.7 ± 3.3 | +7.6 ± 1.4 |

Table D.3: Comparison on Cora-ML of regularly trained models, the state-of-the-art SoftMedian GDC defense, self-trained models and adversarially trained models (train $\epsilon = 20\%$). The first line shows the robust accuracy [%] of a standard GCN and all other numbers represent the difference in robust accuracy achieved by various models in percentage points from this standard GCN. The best model is highlighted in bold.

| Model | Self-trn. | Adv. trn. | A. eval. → A. trn. ↓ | Clean | LR-BCD | PR-BCD | LR-BCD | PR-BCD | LR-BCD | PR-BCD |
|---|---|---|---|---|---|---|---|---|---|---|
| | | | | | $\epsilon$=5% | | $\epsilon$=10% | | $\epsilon$=25% | |
| GCN | ✗ | ✗ | - | 82.5 ± 1.9 | 66.0 ± 2.8 | 66.2 ± 2.5 | 59.2 ± 2.8 | 57.4 ± 2.3 | 48.5 ± 1.6 | 38.0 ± 2.3 |
| GAT | ✗ | ✗ | - | -1.8 ± 0.4 | -5.6 ± 2.1 | -6.3 ± 2.1 | -11.3 ± 2.1 | -6.0 ± 2.2 | -21.4 ± 4.3 | -4.9 ± 2.6 |
| APPNP | ✗ | ✗ | - | +0.6 ± 0.7 | +3.3 ± 1.6 | +2.3 ± 1.9 | +5.3 ± 0.2 | +3.1 ± 0.8 | +7.0 ± 1.1 | +4.3 ± 1.1 |
| GPRGNN | ✗ | ✗ | - | +0.9 ± 0.7 | +5.0 ± 0.8 | +2.5 ± 0.4 | +5.6 ± 0.6 | +4.5 ± 0.5 | +8.6 ± 0.1 | +8.6 ± 1.4 |
| ChebNetII | ✗ | ✗ | - | -0.2 ± 0.5 | +4.2 ± 1.7 | +1.4 ± 0.7 | +6.0 ± 1.3 | +2.6 ± 0.2 | +7.0 ± 1.5 | +6.3 ± 2.1 |
| SoftMedian | ✗ | ✗ | - | -0.8 ± 1.0 | +7.7 ± 1.9 | +4.8 ± 1.7 | +10.7 ± 1.4 | +7.7 ± 0.9 | +17.0 ± 0.6 | +17.4 ± 0.5 |
| GCN | ✓ | ✗ | - | +0.7 ± 0.7 | +4.5 ± 2.0 | +2.6 ± 1.6 | +6.1 ± 2.4 | +3.9 ± 1.3 | +7.3 ± 0.5 | +6.3 ± 0.7 |
| GAT | ✓ | ✗ | - | -3.6 ± 0.5 | -2.8 ± 2.7 | -1.9 ± 3.7 | -6.9 ± 2.2 | -1.3 ± 3.3 | -17.0 ± 3.8 | -2.0 ± 3.1 |
| APPNP | ✓ | ✗ | - | +1.4 ± 0.7 | +5.8 ± 2.0 | +4.6 ± 1.6 | +7.3 ± 1.7 | +5.2 ± 0.4 | +10.1 ± 0.5 | +7.7 ± 1.6 |
| GPRGNN | ✓ | ✗ | - | **+2.1 ± 1.3** | +10.7 ± 1.8 | +7.7 ± 1.8 | +13.5 ± 1.5 | +9.9 ± 2.1 | +17.0 ± 1.4 | +16.9 ± 2.4 |
| ChebNetII | ✓ | ✗ | - | +1.8 ± 0.5 | +8.0 ± 1.7 | +4.9 ± 2.1 | +9.9 ± 2.2 | +6.2 ± 1.8 | +10.8 ± 1.7 | +10.1 ± 2.3 |
| GCN | ✓ | ✓ | LR-BCD | +1.2 ± 0.7 | +6.1 ± 1.9 | +3.5 ± 1.4 | +8.6 ± 1.8 | +3.1 ± 0.5 | +10.9 ± 0.5 | +7.2 ± 1.6 |
| GCN | | | PR-BCD | +0.5 ± 0.6 | +5.4 ± 1.3 | +3.1 ± 1.0 | +8.2 ± 1.1 | +3.5 ± 1.0 | +10.3 ± 0.7 | +6.7 ± 0.7 |
| GAT | ✓ | ✓ | LR-BCD | -2.7 ± 0.6 | +4.3 ± 1.9 | +1.8 ± 1.6 | +7.0 ± 1.9 | +7.2 ± 1.9 | +10.7 ± 0.1 | +16.9 ± 0.6 |
| GAT | | | PR-BCD | -2.1 ± 1.0 | +2.9 ± 2.7 | +1.3 ± 2.2 | +5.0 ± 3.0 | +4.8 ± 2.6 | +5.6 ± 1.4 | +13.0 ± 1.6 |
| APPNP | ✓ | ✓ | LR-BCD | +0.9 ± 0.9 | +6.7 ± 2.1 | +4.7 ± 1.4 | +9.6 ± 1.4 | +5.4 ± 1.2 | +12.9 ± 0.5 | +8.2 ± 1.0 |
| APPNP | | | PR-BCD | +1.1 ± 0.4 | +7.3 ± 1.6 | +5.8 ± 1.7 | +10.6 ± 1.6 | +6.0 ± 0.9 | +13.4 ± 0.9 | +8.6 ± 0.7 |
| GPRGNN | ✓ | ✓ | LR-BCD | +0.8 ± 1.2 | +12.0 ± 3.2 | +8.9 ± 2.7 | +17.6 ± 3.7 | +13.1 ± 2.8 | +25.9 ± 3.4 | +22.8 ± 4.4 |
| GPRGNN | | | PR-BCD | -0.0 ± 1.6 | **+13.4 ± 3.5** | **+10.7 ± 2.9** | **+18.2 ± 4.4** | **+16.1 ± 3.9** | **+28.2 ± 3.6** | **+31.0 ± 4.9** |
| ChebNetII | ✓ | ✓ | LR-BCD | +0.9 ± 0.6 | +7.9 ± 1.6 | +6.2 ± 1.7 | +10.9 ± 1.9 | +8.7 ± 0.8 | +15.0 ± 0.1 | +14.8 ± 1.2 |
| ChebNetII | | | PR-BCD | +1.1 ± 0.4 | +8.5 ± 2.0 | +6.5 ± 1.2 | +13.1 ± 1.7 | +8.9 ± 0.9 | +15.1 ± 1.1 | +15.1 ± 1.1 |

Table D.4: Comparison on Pubmed of regularly trained models, self-trained models and adversarially trained models (train $\epsilon = 20\%$). The first line shows the robust accuracy [%] of a standard GCN and all other numbers represent the difference in robust accuracy achieved by various models in percentage points from this standard GCN. The best model is highlighted in bold.

| Model | Self-trn. | Adv. trn. | A. eval. → A. trn. ↓ | Clean | LR-BCD | PR-BCD | LR-BCD | PR-BCD | LR-BCD | PR-BCD |
|---|---|---|---|---|---|---|---|---|---|---|
| | | | | | $\epsilon$=5% | | $\epsilon$=10% | | $\epsilon$=25% | |
| GCN | ✗ | ✗ | - | 77.4 ± 0.4 | 64.3 ± 0.9 | 62.2 ± 0.8 | 60.4 ± 0.7 | 54.0 ± 0.5 | 52.8 ± 1.4 | 39.3 ± 1.2 |
| GAT | ✗ | ✗ | - | -3.0 ± 1.7 | -5.6 ± 4.5 | -1.5 ± 2.8 | -10.3 ± 6.5 | -0.9 ± 3.7 | -14.8 ± 8.5 | -0.9 ± 6.1 |
| APPNP | ✗ | ✗ | - | +0.5 ± 0.5 | +1.9 ± 0.0 | +1.8 ± 0.9 | +1.7 ± 0.1 | +2.4 ± 0.2 | +2.4 ± 0.2 | +1.1 ± 0.9 |
| GPRGNN | ✗ | ✗ | - | +1.4 ± 1.3 | +1.3 ± 2.0 | +2.8 ± 1.2 | +0.7 ± 2.1 | +3.0 ± 1.7 | +1.2 ± 2.3 | +3.4 ± 1.0 |
| GCN | ✓ | ✗ | - | +1.8 ± 0.9 | +3.4 ± 1.3 | +3.1 ± 1.0 | +4.0 ± 1.5 | +4.1 ± 0.6 | +5.1 ± 1.7 | +2.9 ± 0.6 |
| GAT | ✓ | ✗ | - | -0.3 ± 1.0 | -0.1 ± 1.4 | +0.5 ± 0.3 | -2.3 ± 1.9 | +0.6 ± 1.9 | -3.8 ± 3.4 | -1.7 ± 2.5 |
| APPNP | ✓ | ✗ | - | +2.7 ± 1.1 | +5.5 ± 1.3 | +5.7 ± 1.2 | +6.1 ± 1.2 | +6.7 ± 1.1 | +7.3 ± 1.8 | +6.0 ± 1.3 |
| GPRGNN | ✓ | ✗ | - | +1.7 ± 1.1 | +6.0 ± 1.5 | +5.6 ± 0.9 | +6.6 ± 1.8 | +7.8 ± 1.4 | +9.0 ± 2.9 | +11.5 ± 2.1 |
| GCN | ✓ | ✓ | LR-BCD | +2.2 ± 0.8 | +5.3 ± 1.1 | +4.2 ± 0.9 | +6.1 ± 1.0 | +4.4 ± 0.5 | +7.9 ± 1.4 | +4.5 ± 0.9 |
| GCN | | | PR-BCD | +1.1 ± 1.6 | +5.8 ± 1.2 | +4.3 ± 1.2 | +6.8 ± 0.9 | +5.0 ± 1.2 | +8.8 ± 1.3 | +5.3 ± 0.7 |
| GAT | ✓ | ✓ | LR-BCD | -0.1 ± 1.3 | +4.0 ± 1.0 | +4.7 ± 1.2 | +4.9 ± 0.8 | +8.2 ± 0.4 | +8.4 ± 1.3 | +13.0 ± 1.8 |
| GAT | | | PR-BCD | -0.3 ± 1.0 | +3.2 ± 1.5 | +3.8 ± 1.4 | +3.1 ± 1.3 | +6.2 ± 0.7 | +4.1 ± 1.7 | +9.0 ± 2.5 |
| APPNP | ✓ | ✓ | LR-BCD | +2.6 ± 1.4 | +6.9 ± 1.2 | +6.9 ± 1.4 | +8.5 ± 1.2 | +8.8 ± 1.0 | +11.2 ± 1.7 | +8.8 ± 1.3 |
| APPNP | | | PR-BCD | +2.6 ± 1.3 | +7.8 ± 1.8 | +7.5 ± 1.9 | +9.1 ± 1.5 | +9.4 ± 1.4 | +12.5 ± 2.0 | +9.1 ± 1.9 |
| GPRGNN | ✓ | ✓ | LR-BCD | **+3.1 ± 1.1** | +12.6 ± 1.8 | +9.9 ± 1.1 | +15.4 ± 1.7 | +14.1 ± 1.6 | +21.9 ± 2.5 | +20.6 ± 2.0 |
| GPRGNN | | | PR-BCD | +3.0 ± 0.8 | **+13.6 ± 1.5** | **+13.2 ± 1.0** | **+16.7 ± 1.2** | **+19.0 ± 1.6** | **+23.5 ± 1.9** | **+29.5 ± 1.7** |

Table D.5: Comparison on OGB-arXiv of regularly trained models and adversarially trained models (train $\epsilon = 5\%$). All numbers are in %. The best model is highlighted in bold. In general, adversarial training is effective in increasing adversarial robustness.

| Model | Adv. trn. | A. eval. → A. trn. ↓ | Clean | LR-BCD | PR-BCD | LR-BCD | PR-BCD | LR-BCD | PR-BCD | LR-BCD | PR-BCD |
|---|---|---|---|---|---|---|---|---|---|---|---|
| | | | | $\epsilon$=1% | | $\epsilon$=2% | | $\epsilon$=5% | | $\epsilon$=10% | |
| APPNP | ✗ | - | 70.6 ± 0.2 | 61.6 ± 0.3 | 62.6 ± 0.2 | 58.3 ± 0.4 | 59.0 ± 0.3 | 52.4 ± 0.5 | 52.7 ± 0.6 | 48.4 ± 0.7 | 46.9 ± 1.2 |
| GPRGNN | ✗ | - | **71.9 ± 0.1** | 63.9 ± 0.2 | **64.8 ± 0.2** | 60.7 ± 0.3 | **61.0 ± 0.2** | 54.6 ± 0.4 | 54.0 ± 0.4 | 52.3 ± 0.8 | 50.3 ± 0.8 |
| APPNP | ✓ | LR-BCD | 68.2 ± 0.1 | 63.1 ± 0.1 | 63.2 ± 0.1 | 60.4 ± 0.1 | 60.3 ± 0.1 | **56.5 ± 0.2** | **55.2 ± 0.1** | **54.0 ± 0.1** | **50.4 ± 0.2** |
| APPNP | | PR-BCD | 68.2 ± 0.4 | 62.8 ± 0.3 | 62.5 ± 0.3 | 59.8 ± 0.3 | 59.0 ± 0.3 | 55.4 ± 0.2 | 52.0 ± 0.2 | 52.6 ± 0.3 | 44.7 ± 0.3 |
| GPRGNN | ✓ | LR-BCD | 70.9 ± 0.0 | **64.4 ± 0.1** | 63.8 ± 0.1 | **61.1 ± 0.1** | 59.8 ± 0.1 | 55.9 ± 0.1 | 53.4 ± 0.1 | 53.5 ± 0.1 | 48.4 ± 0.1 |
| GPRGNN | | PR-BCD | 70.3 ± 0.2 | 63.8 ± 0.2 | 63.5 ± 0.2 | 60.6 ± 0.2 | 59.6 ± 0.2 | 55.5 ± 0.2 | 52.0 ± 0.2 | 52.7 ± 0.2 | 44.4 ± 0.1 |

## D.2 Learned Diffusion Coefficients

Figure D.1 shows the learned coefficients $\gamma$ for GPRGNN on Citeseer, Cora, Cora-ML and Pubmed. The results for Citeseer are the ones reported in Figure 6 and given again for reference. The observation that including local constraints during adversarial training leads to a robust model leveraging the

Table D.6: Further adversarial attacks on Citeseer and additional GRAND defense. The LR-BCD results also used LR-BCD during the adversarial training. For PGD and DICE, we perform adversarial training with PR-BCD to match the set of admissible perturbations.

| Attack evaluation | Model | Adv. trn. | Clean | $\epsilon=0.1$ | $\epsilon=0.25$ |
|---|---|---|---|---|---|
| **LR-BCD** | GCN | ✗ | $72.0 \pm 1.6$ | $53.2 \pm 1.9$ | $41.7 \pm 2.8$ |
| | | ✓ | $72.0 \pm 1.5$ | $59.3 \pm 1.5$ | $48.8 \pm 2.4$ |
| | SoftMedian GDC | ✗ | $72.9 \pm 0.8$ | $62.6 \pm 1.0$ | $57.1 \pm 2.1$ |
| | GRAND | ✗ | $74.6 \pm 2.4$ | $65.7 \pm 3.3$ | $59.7 \pm 3.3$ |
| | GPRGNN | ✗ | $74.1 \pm 1.1$ | $58.0 \pm 1.0$ | $49.1 \pm 1.1$ |
| | | ✓ | $72.9 \pm 0.6$ | $67.8 \pm 0.8$ | $64.6 \pm 1.2$ |
| **PGD** | GCN | ✗ | $73.4 \pm 1.3$ | $57.9 \pm 3.0$ | $46.4 \pm 3.1$ |
| | GPRGNN | ✗ | $74.1 \pm 0.2$ | $63.2 \pm 1.3$ | $57.6 \pm 2.8$ |
| | | ✓ | $73.7 \pm 0.6$ | $67.6 \pm 1.2$ | $64.8 \pm 1.3$ |
| **DICE** | GCN | ✗ | $73.4 \pm 1.3$ | $71.8 \pm 1.5$ | $68.8 \pm 1.1$ |
| | GPRGNN | ✗ | $74.1 \pm 0.2$ | $73.8 \pm 1.0$ | $70.9 \pm 0.8$ |
| | | ✓ | $73.7 \pm 0.6$ | $73.7 \pm 1.0$ | $73.1 \pm 0.8$ |

(a) Citeseer

(b) Cora

(c) Cora-ML

(d) Pubmed

Figure D.1: Learned robust diffusion (GPRGNN) coefficients $\gamma$ for different datasets. For each dataset, left figure represents standard training, middle figure represents LR-BCD (adv. training w/ local constraints), right figure PR-BCD (adv. training w/o local constraints). For adversarial training, $\epsilon = 20\%$ is used.

general graph structure in a robust way compared to training with global constraints only is general for all datasets. Training only with global constraints, for node classification, the information in the graph other than the node's own features are suppressed, as only $\gamma_0$ and $\gamma_2$ and, to a lesser extent, $\gamma_4$ are pronounced. These mainly correspond to information about the original node itself, as the information about itself propagates to its neighbor and back in 2-hops and similar for the second-hop neighbors. However, information from $\gamma_k$ with $k$ odd is suppressed through being set close to zero.

A spectral analysis as carried out in Figure 7 can be found for Cora and Cora-ML (and Citeseer for reference) in Figure D.2. The behavior that PR-BCD training leads to high-pass behavior is consistent across datasets.

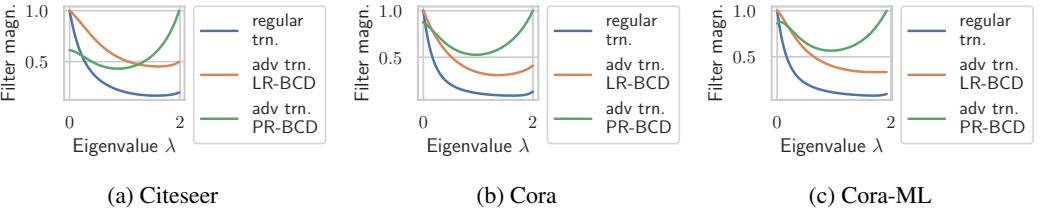

|       (a) Citeseer       |       (b) Cora       |       (c) Cora-ML       |

Figure D.2: Spectral filters corresponding to Figure D.1 for Citeseer, Cora and Cora-ML. Low filter magnitude denotes suppression of the respective frequencies associated with eigenvalue $\lambda$.

## D.3 Comparison of PR-BCD and LR-BCD

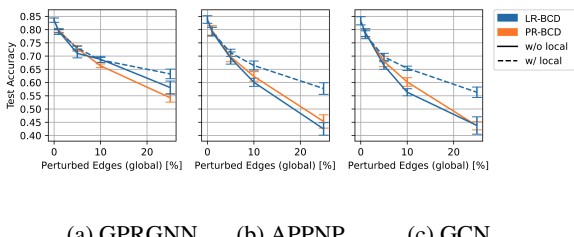

(a) GPRGNN    (b) APPNP    (c) GCN

Figure D.3: Comparison of PR-BCD and LR-BCD on Cora-ML. The x-axis gives the relative global budget. The local budget (if considered) for each node $u$ of degree $d_u$ is $\Delta_u^{(l)} = \lfloor d_u/2 \rfloor$. The models are learned using self-training. Results are aggregated over three different data splits.

Figure D.3 compares the efficacy of PR-BCD to LR-BCD on Cora-ML. For a fair comparison, as the admissible perturbations $\mathcal{B}(\mathcal{G})$ are inherently different for both attacks, we also consider LR-BCD with an unlimited local budget. In this case, LR-BCD's projection defaults to greedily picking the highest-valued perturbations until the global budget is met. We observe that LR-BCD w/o local constraints performs similar to PR-BCD. This suggests that the chosen greedy approach is suitable. Additionally including local constraints, the attack efficacy reduces. Still, even with local constraints, LR-BCD is able to recover adversarial examples that significantly reduce classification performance.

## D.4 Certified Accuracy

Table D.7 shows certified accuracies for Citeseer against an extended set of perturbations compared to Table 1. Table D.8 and Table D.9 show analogous results for the benchmark graphs Cora and Cora-ML, respectively. On all datasets, highest certifiable accuracy is achieved through robust diffusion.

Table D.7: Comparison of certifiable accuracies for different perturbations on Citeseer of regularly trained models, the state-of-the-art SoftMedian GDC defense, and adversarially trained models (train $\epsilon = 20\%$). The first line shows the certifiable accuracy [%] of a standard GCN and all other numbers represent the difference in certifiable accuracy achieved by various models in percentage points from this standard GCN. The best model is highlighted in bold.

| Model | Adv. trn. | Training attack | Clean | 1 add. | 2 add. | 3 add. | 4 add. | 1 del. | 3 del. | 5 del. | 7 del. |
|---|---|---|---|---|---|---|---|---|---|---|---|
| GCN | ✗ | - | $38.3 \pm 7.2$ | $15.6 \pm 3.4$ | $1.9 \pm 0.5$ | $1.7 \pm 0.4$ | $0.0 \pm 0.0$ | $25.2 \pm 5.5$ | $12.0 \pm 2.8$ | $4.8 \pm 1.0$ | $1.9 \pm 0.5$ |
| GAT | ✗ | - | $-14.0 \pm 7.6$ | $-15.1 \pm 3.2$ | $-1.9 \pm 0.5$ | $-1.7 \pm 0.4$ | $+0.0 \pm 0.0$ | $-19.3 \pm 5.2$ | $-12.0 \pm 2.8$ | $-4.8 \pm 1.0$ | $-1.9 \pm 0.5$ |
| APPNP | ✗ | - | $+8.9 \pm 5.8$ | $+26.2 \pm 3.4$ | $+31.8 \pm 4.4$ | $+31.2 \pm 4.0$ | $+26.6 \pm 2.2$ | $+19.0 \pm 3.2$ | $+28.7 \pm 3.4$ | $+31.6 \pm 4.0$ | $+31.9 \pm 4.5$ |
| GPRGNN | ✗ | - | $+17.9 \pm 4.4$ | $+34.6 \pm 2.0$ | $+42.5 \pm 2.7$ | $+42.4 \pm 2.8$ | $+37.5 \pm 2.9$ | $+26.8 \pm 3.0$ | $+36.6 \pm 2.1$ | $+41.3 \pm 2.4$ | $+42.5 \pm 2.7$ |
| ChebNetII | ✗ | - | $+24.6 \pm 9.9$ | $+45.5 \pm 4.4$ | $+55.9 \pm 1.3$ | $+55.6 \pm 1.2$ | $+54.8 \pm 2.0$ | $+36.9 \pm 7.3$ | $+47.8 \pm 3.2$ | $+54.0 \pm 0.9$ | $+56.1 \pm 1.2$ |
| SoftMedian | ✗ | - | $+25.2 \pm 10.5$ | $+47.4 \pm 4.0$ | $+60.1 \pm 1.3$ | $+60.3 \pm 1.4$ | $+61.5 \pm 2.0$ | $+38.2 \pm 7.6$ | $+50.6 \pm 3.0$ | $+57.5 \pm 0.8$ | $+60.3 \pm 1.4$ |
| GCN | ✓ | LR-BCD | $-3.0 \pm 6.0$ | $+11.2 \pm 3.3$ | $+11.4 \pm 2.9$ | $+10.7 \pm 2.9$ | $+6.1 \pm 1.7$ | $+5.1 \pm 5.5$ | $+10.3 \pm 3.4$ | $+13.1 \pm 3.2$ | $+11.5 \pm 3.0$ |
| GCN | ✓ | PR-BCD | $+4.2 \pm 9.2$ | $+10.1 \pm 3.0$ | $+10.6 \pm 3.3$ | $+10.1 \pm 3.5$ | $+6.2 \pm 2.8$ | $+6.9 \pm 6.3$ | $+10.7 \pm 2.0$ | $+11.4 \pm 2.8$ | $+10.6 \pm 3.4$ |
| GAT | ✓ | LR-BCD | $-2.0 \pm 14.6$ | $+5.1 \pm 5.9$ | $+2.6 \pm 1.1$ | $+1.4 \pm 1.1$ | $+1.6 \pm 0.7$ | $+3.0 \pm 10.7$ | $+4.2 \pm 4.4$ | $+4.0 \pm 1.3$ | $+3.0 \pm 1.0$ |
| GAT | ✓ | PR-BCD | $-10.1 \pm 8.7$ | $+5.0 \pm 2.5$ | $+1.9 \pm 1.1$ | $+1.2 \pm 1.3$ | $+2.2 \pm 0.8$ | $-0.8 \pm 5.5$ | $+4.5 \pm 1.3$ | $+2.8 \pm 0.8$ | $+2.2 \pm 0.9$ |
| APPNP | ✓ | LR-BCD | $+21.7 \pm 6.0$ | $+38.0 \pm 1.9$ | $+41.6 \pm 1.9$ | $+41.3 \pm 1.8$ | $+36.6 \pm 1.8$ | $+31.2 \pm 4.0$ | $+40.2 \pm 1.5$ | $+44.1 \pm 0.8$ | $+41.9 \pm 2.0$ |
| APPNP | ✓ | PR-BCD | $+19.3 \pm 4.6$ | $+37.5 \pm 1.2$ | $+41.0 \pm 2.4$ | $+41.0 \pm 2.4$ | $+37.9 \pm 2.6$ | $+29.8 \pm 2.4$ | $+38.9 \pm 1.5$ | $+41.9 \pm 1.9$ | $+41.4 \pm 2.2$ |
| GPRGNN | ✓ | LR-BCD | $\mathbf{+34.0 \pm 7.3}$ | $\mathbf{+55.5 \pm 3.9}$ | $+67.6 \pm 1.1$ | $+67.4 \pm 0.9$ | $+67.9 \pm 0.5$ | $\mathbf{+46.7 \pm 5.8}$ | $+58.6 \pm 3.4$ | $+65.0 \pm 1.6$ | $+67.6 \pm 1.1$ |
| GPRGNN | ✓ | PR-BCD | $+32.9 \pm 7.5$ | $+55.3 \pm 3.9$ | $\mathbf{+68.8 \pm 1.0}$ | $\mathbf{+69.0 \pm 0.9}$ | $\mathbf{+70.4 \pm 0.4}$ | $+45.8 \pm 5.9$ | $\mathbf{+58.9 \pm 3.3}$ | $\mathbf{+65.9 \pm 1.5}$ | $\mathbf{+68.8 \pm 1.0}$ |
| ChebNetII | ✓ | LR-BCD | $+31.5 \pm 12.7$ | $+51.2 \pm 6.5$ | $+62.1 \pm 1.9$ | $+62.3 \pm 1.9$ | $+61.8 \pm 2.2$ | $+43.1 \pm 9.8$ | $+53.9 \pm 5.4$ | $+59.8 \pm 2.5$ | $+62.1 \pm 1.9$ |
| ChebNetII | ✓ | PR-BCD | $+30.7 \pm 13.2$ | $+52.5 \pm 7.0$ | $+65.4 \pm 2.7$ | $+65.4 \pm 2.6$ | $+66.0 \pm 2.1$ | $+43.1 \pm 10.4$ | $+56.1 \pm 5.9$ | $+62.9 \pm 3.7$ | $+65.4 \pm 2.7$ |

Table D.8: Comparison of certifiable accuracies for different perturbations on Cora of regularly trained models, the state-of-the-art SoftMedian GDC defense, and adversarially trained models (train $\epsilon = 20\%$). The first line shows the certifiable accuracy [%] of a standard GCN and all other numbers represent the difference in certifiable accuracy achieved by various models in percentage points from this standard GCN. The best model is highlighted in bold.

| Model | Adv. trn. | Training attack | Certifiable accuracy / sparse smoothing | | | | | | | | |
|---|---|---|---|---|---|---|---|---|---|---|---|
| | | | Clean | 1 add. | 2 add. | 3 add. | 4 add. | 1 del. | 3 del. | 5 del. | 7 del. |
| GCN | ✗ | - | 30.9 ± 0.2 | 19.9 ± 3.2 | 12.9 ± 4.1 | 12.3 ± 4.1 | 5.9 ± 1.7 | 23.7 ± 2.0 | 18.4 ± 3.7 | 15.5 ± 4.1 | 13.2 ± 4.2 |
| GAT | ✗ | - | -0.5 ± 4.8 | -7.8 ± 8.0 | -12.1 ± 3.8 | -11.5 ± 3.8 | -5.5 ± 1.5 | -8.4 ± 6.4 | -11.4 ± 5.5 | -13.9 ± 4.0 | -12.3 ± 4.0 |
| APPNP | ✗ | - | +26.4 ± 1.2 | +30.4 ± 3.5 | +29.5 ± 5.2 | +28.9 ± 5.3 | +30.8 ± 2.5 | +29.4 ± 2.2 | +30.5 ± 4.1 | +28.8 ± 5.2 | +29.5 ± 5.3 |
| GPRGNN | ✗ | - | +20.6 ± 2.4 | +25.6 ± 6.1 | +25.3 ± 6.3 | +25.4 ± 6.1 | +27.0 ± 4.0 | +24.2 ± 4.9 | +26.3 ± 6.2 | +25.4 ± 6.2 | +25.2 ± 6.4 |
| ChebNetII | ✗ | - | +32.8 ± 1.8 | +40.7 ± 6.9 | +46.2 ± 7.8 | +46.8 ± 7.8 | +52.4 ± 4.2 | +38.1 ± 5.2 | +42.1 ± 7.6 | +44.7 ± 8.2 | +46.2 ± 8.1 |
| SoftMedian | ✗ | - | +36.0 ± 1.2 | +46.2 ± 4.6 | +51.5 ± 5.9 | +51.9 ± 5.8 | +57.4 ± 3.0 | +42.5 ± 2.9 | +47.0 ± 5.3 | +49.2 ± 5.7 | +51.4 ± 6.2 |
| GCN | ✓ | LR-BCD | +4.5 ± 2.5 | +5.6 ± 3.6 | -1.3 ± 6.6 | -1.7 ± 6.7 | -2.3 ± 3.3 | +6.3 ± 2.0 | +3.8 ± 4.9 | +0.7 ± 6.1 | -1.2 ± 6.9 |
| GCN | ✓ | PR-BCD | +13.7 ± 6.2 | +7.9 ± 4.5 | -2.0 ± 1.0 | -3.3 ± 1.5 | -2.7 ± 0.7 | +10.6 ± 5.1 | +4.8 ± 3.0 | +0.0 ± 0.6 | -2.0 ± 1.2 |
| GAT | ✓ | LR-BCD | +0.6 ± 0.5 | +9.4 ± 3.1 | +12.2 ± 3.5 | +12.1 ± 3.8 | +3.4 ± 3.1 | +6.2 ± 2.0 | +10.7 ± 3.5 | +12.1 ± 3.7 | +12.0 ± 3.7 |
| GAT | ✓ | PR-BCD | +6.4 ± 1.4 | +1.5 ± 7.4 | -4.8 ± 1.5 | -4.2 ± 1.6 | -2.4 ± 1.0 | +4.8 ± 4.7 | -2.9 ± 4.2 | -5.7 ± 2.0 | -4.6 ± 1.6 |
| APPNP | ✓ | LR-BCD | +25.3 ± 3.5 | +31.1 ± 6.6 | +30.2 ± 6.9 | +30.2 ± 6.9 | +32.5 ± 4.0 | +28.7 ± 5.5 | +30.8 ± 7.0 | +30.4 ± 7.3 | +29.9 ± 7.1 |
| APPNP | ✓ | PR-BCD | +31.5 ± 1.1 | +36.4 ± 3.3 | +36.6 ± 4.6 | +35.8 ± 4.5 | +37.4 ± 2.2 | +35.4 ± 1.7 | +37.1 ± 3.7 | +36.9 ± 4.1 | +36.6 ± 4.7 |
| GPRGNN | ✓ | LR-BCD | +43.8 ± 0.5 | +51.4 ± 3.4 | +54.6 ± 4.9 | +55.2 ± 4.9 | +59.1 ± 3.0 | +49.1 ± 2.3 | +52.5 ± 3.8 | +53.8 ± 4.5 | +54.3 ± 5.1 |
| GPRGNN | ✓ | PR-BCD | **+45.5 ± 0.7** | **+55.7 ± 2.8** | **+61.9 ± 3.6** | **+62.5 ± 3.5** | **+68.6 ± 1.3** | **+52.1 ± 1.6** | **+56.8 ± 3.1** | **+59.5 ± 3.4** | **+61.7 ± 3.7** |
| ChebNetII | ✓ | LR-BCD | +24.8 ± 2.8 | +34.1 ± 4.5 | +38.5 ± 5.3 | +38.7 ± 5.5 | +41.8 ± 1.7 | +30.9 ± 3.1 | +35.2 ± 5.2 | +36.5 ± 5.3 | +38.3 ± 5.5 |
| ChebNetII | ✓ | PR-BCD | +27.1 ± 2.0 | +35.8 ± 6.1 | +38.6 ± 7.0 | +39.1 ± 7.0 | +44.7 ± 3.3 | +33.9 ± 3.6 | +36.5 ± 6.8 | +38.3 ± 7.7 | +38.3 ± 7.2 |

Table D.9: Comparison of certifiable accuracies for different perturbations on Cora-ML of regularly trained models, the state-of-the-art SoftMedian GDC defense, and adversarially trained models (train $\epsilon = 20\%$). The first line shows the certifiable accuracy [%] of a standard GCN and all other numbers represent the difference in certifiable accuracy achieved by various models in percentage points from this standard GCN. The best model is highlighted in bold.

| Model | Adv. trn. | Training attack | Certifiable accuracy / sparse smoothing | | | | | | | | |
|---|---|---|---|---|---|---|---|---|---|---|---|
| | | | Clean | 1 add. | 2 add. | 3 add. | 4 add. | 1 del. | 3 del. | 5 del. | 7 del. |
| GCN | ✗ | - | 37.6 ± 0.9 | 23.6 ± 2.1 | 9.6 ± 3.8 | 8.8 ± 3.7 | 3.8 ± 2.2 | 29.9 ± 0.8 | 19.7 ± 2.9 | 14.3 ± 4.1 | 10.1 ± 3.8 |
| GAT | ✗ | - | -12.4 ± 4.1 | -3.6 ± 4.4 | -2.8 ± 6.0 | -4.2 ± 5.3 | -3.3 ± 2.3 | -7.6 ± 3.8 | -0.4 ± 4.8 | -2.7 ± 7.5 | -3.2 ± 6.1 |
| APPNP | ✗ | - | +17.3 ± 1.5 | +26.1 ± 1.3 | +34.5 ± 3.5 | +33.8 ± 3.3 | +34.9 ± 2.0 | +21.9 ± 0.4 | +29.0 ± 2.1 | +32.3 ± 3.5 | +34.0 ± 3.5 |
| GPRGNN | ✗ | - | +25.4 ± 3.4 | +32.0 ± 1.8 | +38.1 ± 2.9 | +38.6 ± 2.8 | +39.3 ± 1.7 | +29.2 ± 2.2 | +34.5 ± 1.7 | +35.6 ± 3.0 | +37.8 ± 2.9 |
| ChebNetII | ✗ | - | +29.9 ± 3.2 | +40.8 ± 6.6 | +51.6 ± 9.2 | +52.5 ± 9.1 | +55.3 ± 7.1 | +35.9 ± 4.7 | +43.8 ± 7.5 | +48.0 ± 9.0 | +51.4 ± 9.2 |
| SoftMedian | ✗ | - | +35.6 ± 0.4 | +46.7 ± 5.1 | +57.2 ± 7.3 | +57.7 ± 7.5 | +60.8 ± 5.0 | +41.8 ± 2.8 | +49.8 ± 6.1 | +53.8 ± 7.9 | +56.9 ± 7.4 |
| GCN | ✓ | LR-BCD | +11.9 ± 4.3 | +7.6 ± 1.9 | +1.5 ± 5.4 | +0.7 ± 5.5 | +0.1 ± 3.5 | +8.2 ± 1.6 | +6.6 ± 3.4 | +3.5 ± 6.4 | +1.4 ± 5.6 |
| GCN | ✓ | PR-BCD | +1.8 ± 3.8 | +1.2 ± 4.3 | +2.8 ± 7.1 | +2.8 ± 7.1 | +3.5 ± 5.1 | +0.0 ± 3.6 | +1.3 ± 5.3 | +2.0 ± 6.9 | +2.8 ± 7.3 |
| GAT | ✓ | LR-BCD | -5.5 ± 3.1 | +5.6 ± 1.4 | +11.7 ± 7.4 | +11.6 ± 7.7 | +10.2 ± 6.7 | +0.6 ± 0.7 | +8.3 ± 2.8 | +10.8 ± 5.8 | +11.5 ± 7.4 |
| GAT | ✓ | PR-BCD | -13.0 ± 1.6 | -3.3 ± 6.6 | +6.7 ± 10.4 | +7.4 ± 10.4 | +10.0 ± 8.6 | -8.1 ± 4.3 | -0.9 ± 7.4 | +2.1 ± 9.6 | +6.2 ± 10.5 |
| APPNP | ✓ | LR-BCD | +28.4 ± 0.5 | +34.3 ± 2.9 | +39.1 ± 4.5 | +39.0 ± 4.3 | +39.4 ± 3.0 | +30.9 ± 1.8 | +36.0 ± 3.4 | +37.2 ± 4.8 | +39.0 ± 4.5 |
| APPNP | ✓ | PR-BCD | +25.6 ± 1.9 | +32.7 ± 3.8 | +38.5 ± 5.2 | +38.6 ± 5.1 | +38.8 ± 3.9 | +34.5 ± 4.2 | +36.3 ± 5.0 | +38.0 ± 5.3 | |
| GPRGNN | ✓ | LR-BCD | +41.2 ± 1.4 | +52.9 ± 1.8 | +64.9 ± 3.2 | +65.3 ± 3.0 | +68.7 ± 1.8 | +47.8 ± 0.8 | +56.5 ± 2.6 | +61.0 ± 3.7 | +64.4 ± 3.2 |
| GPRGNN | ✓ | PR-BCD | **+42.4 ± 1.6** | **+56.0 ± 1.6** | **+69.5 ± 3.2** | **+70.2 ± 3.2** | **+74.9 ± 1.8** | **+49.9 ± 0.6** | **+59.6 ± 2.5** | **+65.0 ± 3.7** | **+69.0 ± 3.3** |
| ChebNetII | ✓ | LR-BCD | +39.6 ± 1.9 | +51.4 ± 2.7 | +62.1 ± 5.7 | +62.8 ± 5.5 | +65.7 ± 3.5 | +45.7 ± 0.8 | +54.6 ± 3.9 | +58.7 ± 6.1 | +61.6 ± 5.8 |
| ChebNetII | ✓ | PR-BCD | +36.7 ± 1.2 | +49.8 ± 3.8 | +62.2 ± 6.5 | +63.0 ± 6.4 | +66.2 ± 3.9 | +43.9 ± 1.5 | +53.2 ± 5.1 | +58.2 ± 6.9 | +61.7 ± 6.6 |

# E   Further Related Work

**Defenses on Graphs.** Next to adversarial training, previous works proposed defenses based on smoothing [32], preprocessing the graph structure [50–52], or using robust model structures [46, 48]. For an extended survey, we refer to Günnemann [12].

**Attacks on Graphs.** Generally, one distinguishes evasion attacks, perturbing the graph after training [7, 2, 17] and poisoning attacks, perturbing the graph before training [7, 47]. Further, one differentiates global attacks perturbing multiple node predictions at once [17] from local (targeted) attacks perturbing a single node's prediction [7]. Attacks can perturb the nodes attributes [7], the edge structure [7, 47, 49, 2, 17], or insert malicious nodes [53]. An adversarial attack should not change the true classes (unnoticeability) [18]. For certain applications, like combinatorial optimization, it might be known how the true label is affected by perturbation. For example, Geisler et al. [54] study such adversarial attacks (& training) for GNNs for combinatorial optimization. However, in the context of node classification, this relationship is fuzzier. For example, Zügner et al. [7] define an attack that preserves the global degree distribution, or Gosch et al. [6] analyzes semantic preservation in synthetic graphs and show that changing more edges than the degree of nodes is usually never unnoticeable. In this work, we focus on *evasion* attacks on the edge structure, aiming at *globally* attacking node predictions with realistic *local constraints* enforcing unnoticeability.

# F    Limitations

This work focuses on adversarial robustness against structure perturbations for node classification. While this is the most studied and prevalent case in previous works (see Günnemann [12] and Table 3), there are many other interesting graph learning tasks or extended settings, out of scope for this work, for which robustness and adversarial training schemes could be of interest, as e.g., link prediction, node feature perturbations or graph injection. Conceptually, the adversarial training procedure and LR-BCD attack can also be applied to the aforementioned tasks as long as the model remains differentiable.

On another note, while we finally put into the hands of practitioners a global attack method aware of local constraints, there is no clear strategy on how to determine the optimal local budget. Indeed, these have to be set application dependent by domain experts. For example, Gosch et al. [6] leverage knowledge about the data generating distribution to determine budgets that correspond to semantic-preserving perturbations. However, in most real-world applications, we lack knowledge about the data generating distribution. As a result, we also use the different interpretability aspects of robust diffusion to collect additional evidence and insights into the learned message passing (see Section 3 & 5).

In Section 4, we report the asymptotic complexity of our LR-BCD attack. In Section 5, we detail the used hardware for the adversarial training experiments and report the computational cost in terms of runtime as well as memory on the arXiv graph, consisting of 170k nodes.

