# OpenReview forum: "Adversarial Training for Graph Neural Networks: Pitfalls, Solutions, and New Directions"
_NeurIPS.cc/2023/Conference — NeurIPS 2023 poster_

### Official Review · Reviewer_bw8r · 2023-07-05

**Soundness:** 3 good
**Presentation:** 3 good
**Contribution:** 2 fair
**Rating:** 5
**Confidence:** 4

**Summary:**

This paper presents a comprehensive analysis of the transductive setting, highlighting the limitations of achieving perfect robustness solely by memorizing the clean graph. The authors argue that the observed improvements in robustness from previous adversarial training methods stem from unintentional leakage of information about the clean test nodes. To address this issue, the authors explore the inductive setting, which mitigates these drawbacks. They propose a defense mechanism by combining GPRGNN and ChebNetⅡ with adversarial training, effectively countering both globally and locally constrained attacks. For the implementation of the locally constrained attack, the authors develop an approximate solution to a relaxed multi-dimensional knapsack problem. The experimental results presented in the paper showcase the efficacy of the proposed attack and defenses.

**Strengths:**

1.	It is meaningful to discuss the adversarial training on graph models, as this is still an open problem.
2.	I highly value the investigation conducted in the section dedicated to locally constrained attacks, and I find the implementation of LR-BCD to be inherently rational.
3.	Experiments are sufficient enough to demonstrate the efficacy of the proposed methods.



**Weaknesses:**

1.	The primary concern with this work lies in the disparity between the initial analysis and the subsequent defense method design. Although the analysis extensively examines the limitations of transductive settings, there is a noticeable absence of a correlation with the utilization of an adaptive spectral GNN to enhance robustness. The motivation of the proposed defense is not very clear, and it is a combination of existing methods. While adaptively assigning weights to local neighborhoods can be an effective strategy for improving robustness, it appears to be unrelated to the previous discussions on the flaws of transductive settings and adversarial training.

2.	The authors claim that GPRGNN offers interpretability, but Section 3 primarily focuses on presenting the learned message passing weights and filter kernels of GPRGNN. While this aligns with the intuition of enhancing GNN robustness by incorporating high-frequency information and higher-order neighborhoods, it remains challenging to argue that this alone qualifies as interpretability.

3.	The discussion of graph adversarial attacks in the transductive setting remains valuable. On one hand, many methods in the transductive setting do not involve leakage of clean local structures of test nodes, and conventional adversarial training does not include a self-training module. On the other hand, in real-world scenarios, memorizing the graph is not certainly feasible because the structure and labels of graph nodes can be dynamically changing. In fact, even without the addition of new nodes to the graph, the defense cannot classify the current nodes based on the remembered graph structure. For example, in graph-based spammer or fraud detection, benign users and fraudulent users can switch roles as time goes on.

4.	The theoretical part of this work, Section 2.1.1 and Appendix C.1, seems to be needless and trivial. It is evident that in evasion attacks, accuracy will not be affected if clean graphs are used for testing.



**Questions:**

1.	On homophilous graphs, a high-pass filter is likely to decrease accuracy. Figure 8 indicates that adversarial training transforms the GNN into a high-pass filter. However, why does this not result in a trade-off between accuracy and robustness? Can you provide some possible analyses?


2.	Can the authors provide more examples of information leakage in the transductive setting? As far as I know, this is not common.

Overall, this paper is intriguing and attempts to address some of the key challenges in graph adversarial attacks. If the authors address my concerns well, I am willing to consider raising the score.


**Limitations:**

yes

---

> ### Author Rebuttal · Authors · 2023-08-09
>
> We want to thank the reviewer for the constructive review!
>
> ## W1: Relationship of transductive and inductive learning with robust graph diffusion
>
> The use of adaptive spectral GNNs is **directly connected** to the discussion on the flaw of the transductive setting. With our refined model choice, we achieve *perfect robustness* in the transductive setting (see §2.1.2) and make this flaw apparent. Revealing the evaluation flaw and solving the setting stands in stark contrast to prior work and are important contributions on their own (see also our answer to W3).
>
> However, the main motivation for our model choice is the ability to approximate any spectral graph filter. Thus, adversarial training may choose the most robust graph filter that achieves a competitive training loss. Our empirical evaluation shows that such a flexible architecture is key to achieving state-of-the-art robustness via adversarial training (also in the appropriate inductive setting). We make this point in the introduction but will make it and its connection to the evaluation flaw (§2) more prominent in a revised version.
>
> ## W2: Interpretability of robust graph diffuison
>
> In the seminal work [L], Murdoch et al.
> > define interpretable machine learning as the **extraction of relevant knowledge** from a machine-learning model concerning relationships either contained in data or **learned by the model**. Here, we view knowledge as being relevant **if it provides insight for a particular audience into a chosen problem**.
>
> We provide three perspectives that meet this definition, i.e., provide valuable insights into the learned characteristics for machine learning researchers or practitioners. We will clarify in the revised revision that this is what we mean by interpretability.
>
> The (i) polynomial and (ii) spectral perspective are examining a global property of the learned model. Thus, in contrast to, e.g., a GCN, it is much easier to understand how the robust graph diffusion achieves the enhanced robustness.
>
> Moreover, the (iii) spatial perspective also yields local explanations and is more closely related to GNN explanation methods like GNNExplainer [M]. For example, the $v$-th row $\mathbf{T}_v$ of the diffusion matrix $\mathbf{T}$ (see line 232) will yield the influence of the transformed node features of $u \in \\{1, 2, \dots, n\\}$ on the prediction for node $v$ (see also Figure 4).
>
> ## W3: Differences and importance of transductive & inductive learning
>
> > The discussion of graph adversarial attacks in the transductive setting remains valuable. [...] in real-world scenarios, [...] can be dynamically changing. [...]
>
> This is a great point highlighting why our work is crucial for the field. We agree that robustness in the transductive setting is important. However, in combination with an evasion threat model, one needs to be cautious as it poses the risk that "a defense" exploits the availability of the clean test nodes (and their structure) at training time as happened in prior work on adversarial training [A, O]. This can yield a perfectly robust, yet not useful model.
>
> We also agree about dynamic changes in many real-world scenarios. However, this *does not fit* the definition of transductive learning (see §2.1 and [N]), where the test data has to be known at training time. All changes after training constitute an inductive setting (see §A) and we have discussed exactly the proposed scenario in §A lines 541-543. Being inductive it is not sufficient to memorize the available data as is precisely our point in §2.1.1. (lines 126-128).
>
> We think it is important to sensitize the field to the existent evaluation flaw and advocate for adopting learning settings and benchmark datasets better capturing the relevant robustness scenarios.
>
> ## W4: Motivation for theoretical analysis
>
> Although the theoretical findings are intuitive to grasp once one reviews the learning setting properly, it is directional for the field as the implications are not generally known (nor trivial). This is evident by the fact that all adversarial training works do use this setup (see Table 2). While it might be fine to study transductive evasion as a worst-case setting, where all inductive changes are adversarial, it is problematic to exploit the availability of the clean data at training time as done by Xu et al. [A, O] or [P, S].
>
> ## Q1: Accuracy-robustness tradeoff
>
> Indeed, we observe an accuracy-robustness tradeoff. E.g., a regularly trained GPRGNN on Citeseer (Table 1) has the best clean accuracy ($74.2$%) and is a low pass, followed by adv. training with LR-BCD ($73.7$%), which is still a low pass but allows for more high-frequency components. GPRGNN trained with PR-BCD mainly is a high-pass filter allowing low-frequencies only to pass damped achieving the lowest clean accuracy ($72.6$%). Similar behavior can be observed on our other datasets (see §D.1). This can be explained as the adversarial connections between dissimilar (different class) nodes distort homophily and make high-frequency information more important for robust generalization. While having only a global constraint (PR-BCD) allows for an excessive amount of adversarial edges per node (see Figure 5), local constraints (LR-BCD) mainly preserve the optimal filter characteristics.
>
> ## Q2: Information leakage
>
> We focus on adv. training for graph learning, here the first and so far most cited work [A] suffers from the mentioned evaluation flaw as well as their follow-up work [O]. In this setup, the predictions on the clean test nodes are leaked by using self-training (see §2.1.2). For many other defenses it is not clear to what extent the information leakage is exploited. However, there are works where the information leakage might play a role, like in the contemporary work on a causal defense [S] (Fig. 3 shows that the clean/original graph is used during training) or VAT [P]. To avoid reporting biased robustness scores, it would be important to study evasion defenses inductively.

---

> > ### Comment · Reviewer_bw8r · 2023-08-17
> > **Thanks for the rebuttal**
> >
> > I thank the authors for the reply. Most of my concerns are well addressed, and I raise my score. I sincerely suggest that the author include the answer to the first weakness in the paper.
> >
> > However,  I still think the theoretical part (Section 2.1.1 and Appendix C1) is redundant. It is valuable to point out that there can be information leakage when using clean structure of test nodes during adversarial training. But reasoning on a clean, memorized graph is obviously not affected by perturbations and does not require a mathematical form to illustrate this point.

---

> > > ### Author Response · Authors · 2023-08-19
> > > **Thanks for the reply**
> > >
> > > We want to thank the reviewer for the positive reply and for raising the score!
> > >
> > > ## I: Including the answer to W1 in the paper
> > >
> > > >I sincerely suggest that the author include the answer to the first weakness in the paper.
> > >
> > > We plan to include our answer to W1 in the revised paper and hence, better incorporate the motivation for our model choice and its link to the discussion in §2. In particular, we plan the following changes highlighted in **bold** to the paper to make these points more clear:
> > >
> > > 1) In the *GNN architecture* paragraph in the Introduction, we plan to expand on the statement "approximate any spectral graph filter" starting at line 34/35 and adapt the text as follows:
> > >
> > >     >*(ii) GNN Architectures:* [...] Instead, we propose to use flexible message-passing schemes based on learnable diffusion. **The main motivation behind this choice is the ability** to approximate any spectral graph filter. **Thus, adversarial training may choose the most robust filter that achieves a competitive training loss.** **Thereby, we** significantly improve robustness compared to previous work, while yielding an interpretable message-passing scheme, **and making the evaluation bias in the transductive setting (i) apparent.**
> > >
> > > 2) In Section 2.1.2, we plan to adapt the sentence on lines 162/163 as follows:
> > >     > [...] In other words, it [GPRGNN] finds a trivial solution to this learning setting, **making its limitations evident** similar to the theoretic solution discussed in Section 2.1.1. [...]
> > >
> > > 3) In Section 2.1.2, we plan to exchange the last two sentences on lines 169-172 with the following text (bold writing omitted):
> > >
> > >     > We find that flexible architectures like GPRGNN combined with adversarial training (see robust diffusion in subsequent Section 3) not only solve the transductive setting to perfection showcasing their adaptability, but also learn to robustly generalize to unseen nodes far better than previously studied models - while considering an unbiased inductive setting (see Section 5).
> > >
> > > 4) In the beginning of Section 3 *Robust Diffusion*, we plan to adapt the first two sentences and swap their order as follows:
> > >
> > >     > We propose to use learnable graph diffusion models **able to approximate any graph filter** in conjunction with adversarial training to obtain a *robust diffusion*. **The key motivation is to** use more flexible GNN architectures for adversarial training than previous studies. [...]
> > >
> > >  We are looking forward to further suggestions, if the proposed changes do not fully address the reviewer's point.
> > >
> > > ## II: On Section 2.1.1
> > >
> > > >However, I still think the theoretical part (Section 2.1.1 and Appendix C1) is redundant. It is valuable to point out that there can be information leakage when using clean structure of test nodes during adversarial training. But reasoning on a clean, memorized graph is obviously not affected by perturbations and does not require a mathematical form to illustrate this point.
> > >
> > > We want to thank the reviewer for recognizing the value of pointing out the information leakage in Section 2.1.1. We took a lot of effort to present the conceptual differences between the inductive and transductive learning setting in Section 2.1 and 2.1.1 in a way, to make the pitfalls for robustness evaluations as intuitively clear as possible. While it is evident that a memorized graph is not affected by perturbations, the mathematical form serves other purposes:
> > >
> > > * Concerning adversarial training, it showcases how memorization is a valid and optimal solution to the so far studied saddle-point problem arising in the related adversarial training works [A,O]. Thus, it provides important intuition for the *perfect robustness* result presented in Section 2.1.2.
> > > * It further provides intuition, as to why information leakage is **not** an issue in the inductive setting.
> > >
> > > Thus, we think §2.1.1 is important to aid the conceptual shift from a transductive towards an inductive setting. This shift is crucial for further progress in the field of adversarial training for graph learning (and relevant for the graph robustness field in general). For a revised version of §2.1.1, we plan to better work out (and focus) these intuitions and lessen emphasis on Proposition 1.

---

### Official Review · Reviewer_fJD8 · 2023-07-05

**Soundness:** 3 good
**Presentation:** 3 good
**Contribution:** 2 fair
**Rating:** 7
**Confidence:** 4

**Summary:**

This paper shows fundamental limitations of transductive adversarial training of GNNs for node classification and advocates that adversarial training of GNNs should be in an inductive setting. Therefore, this paper proposes a more flexible message-passing scheme using learnable diffusion to gain robust representations under adversarial training, and uses additional local constraints for structural perturbation and an efficient attack (LR-BCD) that contains both local and global perturbations.

**Strengths:**

This paper studies an important problem on the GNN adversarial robustness and provides some interesting observations and insights on this topic. In particular, the analysis and discussions of theoretical and empirical limitations of adversarial training in GNN transductive learning are insightful and useful for the community.

The proposed robust diffusion framework for adversarial training appears efficient in a fully inductive GNN setting with flexible GNN architectures and refined local/global topology perturbations.

**Weaknesses:**

1. The technical contributions of the work seem a bit limited. It appears the proposed method is a combination of existing works, although it is a simple yet effective solution. The claimed novelty lies in the handling of the new local constraints with LR-BCD, but it does not seem to be technically different from PR-BCD. In addition, there is a lack of theoretical justification why the new projection imposes sensible global and local constraints.

2. Given the limited technical contributions of the methodologies, I would like to see more extensive experimental evaluations of the proposed framework. More evaluations on other homophilic and heterophilic datasets with more powerful adversarial attacks (e.g., adaptive attacks [1]) under inductive setting should be conducted to demonstrate the universality of the proposed methods. In addition, the comparison with other robust GNN models (e.g., ProGNN [2], GRAND [3], GARNET [4]).

    [1] Felix Mujkanovic, Simon Geisler, Stephan Günnemann, and Aleksandar Bojchevski. Are defenses for graph neural networks robust? NeurIPS’22.
    [2] Wei Jin, Yao Ma, Xiaorui Liu, Xianfeng Tang, Suhang Wang, and Jiliang Tang. Graph structure learning for robust graph neural networks. SIGKDD’22.
    [3] Wenzheng Feng, Jie Zhang, Yuxiao Dong, Yu Han, Huanbo Luan, Qian Xu, Qiang Yang, Evgeny Kharlamov, and Jie Tang. Graph random neural network for semi-supervised learning on graphs. ICML’21.
    [4] Deng et al. GARNET: Reduced-Rank Topology Learning for Robust and Scalable Graph Neural Networks. LoG'22.

3. Some minor points:
* It does not seem the equation (2) correctly reflects the principles of self-training, because theta’ appears predefined in the expression, while it is actually trained from labelled data.
* In Equation-(5), the sum operation in the constraint is not well defined.
* The authors may want to make the title more specific to better reflect the focus of this work, avoiding potential misleading and vagueness.

**Questions:**

1. Please clarify how L/PR-BCD corresponds to adaptive attacks.
2. Please justify how the additional local constraints could preserve graph semantics.

---

> ### Author Rebuttal · Authors · 2023-08-09
>
> We want to thank the reviewer for the constructive feedback and for noting that we study an important problem while giving interesting observations and insights useful for the community.
>
> ## W1 (a): Technical novelty
>
> We appreciate the reviewer recognizing our effective solution to implementing local constraints. LR-BCD *technically* and *non-trivially* extends PR-BCD, as  PR-BCD does extend PGD [A, B]. We show that including local constraints results in a challenging optimization problem (see Eq. 6 and [C]) and we propose an efficient solution algorithm. This projection step is the *main technical challenge* in these algorithms and **neither** PR-BCD **nor** PGD could handle such constraints. Thus, we introduce the *first* global attack with this ability.
>
> We want to stress that local constraints are widely used in the local attacks [E, Q, U] and certificates literature [K, V]. The lack of such a global attack even led to prior work trying to naively and expensively apply local attacks for each node sequentially to generate locally constrained global perturbation [W]. Thus, if it would not have been technically challenging, most certainly, at least one of the global attacks would have included them [A, B, F, G, H, I, J]. Therefore, this is a clear contribution to the field.
>
> Importantly, novelty is not only derived from the new attack. Regarding test-time attacks in a transductive setting as studied by Xu et al. (2018) [A] and a wide range of other robustness works [B,D,E,K] including all works on adversarial training (see §6) - we achieve practically *perfect robustness*. Prior work did not achieve this, they did not even come close. Furthermore, we use our results to empirically and theoretically reveal the evaluation flaw in the transductive setting and overcome it by moving to a more realistic and challenging inductive setting. This is important and directional for the field as the resulting issue was not generally known. In the more realistic inductive setting, we then, without bells and whistles, achieve state-of-the-art robustness via adversarial training.
>
>
> ## W1 (b)/Q2: Sensibility of global and local constraints and semantic preservation.
>
> Our projection can deal with any (local or global) budget constraints of interest for a particular dataset. Concretely, our projection method can deal with arbitrary perturbation sets $\mathcal{B}(A) = \\{\tilde{A}\ \in \\{0,1\\}^{n\times n} |\ \lVert\tilde{A}-A\rVert_0 \le \Delta\ \land\ \lVert\tilde{A}_i-A_i\rVert_0\le\Delta_i^{(l)}\ \text{for all nodes}\ i\\}$, where $A_i$ denotes the $i-$th row of the adjacency $A$ and $\Delta$ & $\Delta_i^{(l)}$ denote the global and local attack budgets. This is the most general form of the most commonly used structure thread model [D]. Handling it was **not** possible before (see above).
>
> The concrete global and local budget choices, modeling semantic preservation, are application specific and possibly derived from domain knowledge. The sensibility of local constraints is evident by their pervasive use in the local attack [E, Q, U] and certificate literature [K, V]. Furthermore, even small global budgets (e.g., 5% of total edges) result in hundreds of adversarial edges, while for most real-world graphs, the average node degree is very small (e.g., 5.68 on Cora-ML). Figure 5 in §4 shows that successfully attacked nodes by PR-BCD usually have more adversarial edge insertions/deletions than original clean edges and hence, can be seen as completely rewired. This is not only intuitively problematic but Gosch et al. (2023) [B] prove on random graph models that this almost certainly changes the semantics (most likely node labels), whereas smaller changes often do not (see their Table 2). Possible degenerate effects of a fatally strong attack that can completely rewire any individual node are emphasized in Figure 4c where GPRGNN trained using PR-BCD learns to entirely disregard the graph structure. However, Figure 4d shows that with local constraints, graph structure is still leveraged but now, for more robust predictions.
>
> ## W2: More datasets, attacks, and new defense
>
> As requested by the reviewer, we now investigate more datasets, baselines, and attacks. We refer to the meta-comment and attached pdf for the additional experimental results, demonstrating the universality of our methods.
>
> Note that we did not include ProGNN or GARNET as they are defenses against poisoning and hence, not applicable to our work.
>
>
> ## W3-1: On Eq. 2
>
> Eq. 2 represents the adversarial training objective after self-training has already occurred. This is explained in lines 80-81, where we define the self-training step as happening before using Eq. 2 and yielding $\theta'$ through training from the labeled data. We will try to make this more prominent in a revised version of the manuscript.
>
> ## W3-2: Sum operation in Eq. 5
>
> We define the sum operation as summing over all elements of the matrix directly below Eq. (5) in line 287 (& line 300 for Eq. 6). We will revise this in the revision of this paper.
>
> ## W3-3: Title
>
> Given the feedback, we consider to alter the title to *Adversarial Training for Graph Neural Networks: Pitfalls, Solutions and New Directions* and are open to suggestions.
>
> ## Q1: L/PR-BCD are adaptive attacks
>
> An adaptive attack is white-box, i.e., has perfect knowledge about the model, the parameters, and the data, including all defensive measures [X]; as is the case with L/PR-BCD. Note that adversarial training only concerns the training phase and is not an active defense during inference. Furthermore, L/PR-BCD - as being gradient-based and fully white-box - may adapt to the adversarially trained weights. In our attached pdf, we now also compare to other attacks to demonstrate the efficacy and adaptiveness of L/PR-BCD. Additionally, we evaluate certified robustness (see Table 1), where we show to significantly increases provable robustness, which is **independent** of any particular attack.

---

> > ### Comment · Reviewer_fJD8 · 2023-08-19
> >
> > I appreciate the authors' responses to my comments, and they appear convincing. Therefore I would like to raise my rating to "accept".
> >
> > BTW, the new title looks much more appealing to me.

---

> > > ### Author Response · Authors · 2023-08-21
> > >
> > > We thank the reviewer for the positive reply and for raising the score! As mentioned, we will change the title to *Adversarial Training for Graph Neural Networks: Pitfalls, Solutions and New Directions* in the revised paper version.

---

### Official Review · Reviewer_zNuu · 2023-07-08

**Soundness:** 3 good
**Presentation:** 3 good
**Contribution:** 3 good
**Rating:** 7
**Confidence:** 3

**Summary:**

This paper focuses on adversarial training techniques for Graph Neural Networks (GNNs) in the context of inductive semi-supervised learning. It addresses the shortcomings of previous approaches in graph learning: perfect robustness is possible for transductive learning setting under evasion attack. The authors propose a more adaptable GNN model equipped with a learnable robust diffusion mechanism, capable of accommodating adversarial perturbations. Furthermore, the paper introduces a novel method for attacking GNNs by perturbing their structures, which consider local constraints in global attack settings. The study demonstrates the efficacy of adversarial training as a defensive strategy for GNNs and offers valuable insights into the field of adversarial learning for GNNs.

**Strengths:**

- The paper provides a comprehensive summary of existing research and knowledge on adversarial training techniques for Graph Neural Networks (GNNs).

- The authors elaborate the limitations of previous approaches in graph learning settings, particularly emphasizing the challenges associated with transductive learning settings for evasion attacks. It also introduces a more realistic fully inductive setting.

- The authors propose a new adversarial training method combining a learnable graph diffusion, with discussion on the interpretability of the learned robust representation. Based on the revealed importance of local constrained, they further proposed an attack method LR-BCD, extending previous PR-BCD attack with a projection incorporating local constraints.

- The paper presents extensive empirical evidence demonstrating the robustness of the proposed adversarial training techniques and the proposed attack method.

- The paper offers interesting insights for the community regarding the role of local constraints in adversarial training for GNNs.


**Weaknesses:**

- Limited to evasion attack, should clarify this in a early stage of presentation
- Better presentation if the authors shorten section 2.1 and section 2.1.1, which introduce commonly-used settings and quite obvious propositions, and add more model framework details and training details in the main text.

Minor issue:
- the abbreviation GPRGNN is quite confusing while just appears with no explanation in the intro section

**Questions:**

- Curious how to apply adversarial training under poisoning attack settings. it is okay if this is out of the scope of this paper.
- How would the adversarial trained model generalize to other types of attacks? Are the proposed robust diffusion model able to defend other types of attacks? E.g., gray-box/black-box attacks, poisoning attacks.

---

> ### Author Rebuttal · Authors · 2023-08-09
>
> We want to thank the reviewer for the positive review!
>
> ## W1: Clarifying the evasion attack scenario in the beginning
>
> >Limited to evasion attack, should clarify this in a early stage of presentation
>
> Thank you for pointing this out! Currently, we clarify this in line 67 (Section 2). In a revised manuscript, we plan to include this information directly in the introduction.
>
> ## W2: Shorten Section 2.1 and 2.1.1
> >Better presentation if the authors shorten section 2.1 and section 2.1.1, which introduce commonly-used settings and quite obvious propositions, and add more model framework details and training details in the main text.
>
> We do think that it is important to formally introduce and distinguish adversarial robustness in the transductive and inductive learning setting. Although the theoretical finding is intuitive to grasp once one reviews the learning settings properly, it is directional for the field as the resulting issue with a flawed evaluation in the transductive setting was not generally known (nor is trivial). This is evident by the fact that (i) the first and so far most cited work on adversarial training for robust graph learning [A] suffers from the aforementioned evaluation flaw as well as their follow-up work [O] and (ii) all adversarial training works use the transductive setting (see Table 2).
>
> ## M1: GPRGNN abbreviation in introduction
> >the abbreviation GPRGNN is quite confusing while just appears with no explanation in the intro section
>
> We thank the reviewer for pointing this out! We will explain the abbreviation in a revised version of the manuscript.
>
>
> ## Q1: Poisoning
>
> >Curious how to apply adversarial training under poisoning attack settings. it is okay if this is out of the scope of this paper.
>
> Adversarial training is foremost a defense against evasion threat models and is usually not studied in combination with poisoning. Thus, we see poisoning as out of scope for this work. However, recently, there have been a few works (not in the graph domain), investigating the effects of adversarial training given poisoned data [R]. Thus, we see this as a very interesting direction for future work.
>
> ## Q2: Other attacks
>
> >How would the adversarial trained model generalize to other types of attacks? E.g., gray-box/black-box attacks, poisoning attacks.
>
> Note that gray-box or black-box attacks are usually weaker than white-box attacks (i.e., attacks having complete knowledge of the data / model / weights). As one is usually most interested in the worst-case performance of a model [X], we foremost study the white-box attacks L/PR-BCD. However, for the rebuttal, we have now **added two additional attacks**: the black-box attack DICE [Q], and the white-box attack PGD [A]. You can find their performance in the pdf accompanying the rebuttal.
>
> Moreover, we study the generalization to weaker and stronger perturbations throughout our experiments by varying the attack budgets (see, e.g., $\epsilon$ in Table 1). Furthermore, we compare perturbation models w/ and w/o local constraints as well as cross-evaluations using one for adversarial training and the other for test-time evaluation in Table 1 (and §D.1). We also assess the impact on certifiable robustness via randomized smoothing in Table 1 (and §D.1). As argued above, poisoning attacks are out of scope for this work.

---

> > ### Comment · Reviewer_zNuu · 2023-08-21
> > **Thank you for your response**
> >
> > Thank you for your detailed response. My concerns are well addressed. Overall I think the paper is of good quality and I will keep recommending the acceptance of the paper.

---

> > > ### Author Response · Authors · 2023-08-21
> > >
> > > We want to thank the reviewer for acknowledging our rebuttal and the positive reply!

---

### Official Review · Reviewer_ZTec · 2023-07-11

**Soundness:** 3 good
**Presentation:** 4 excellent
**Contribution:** 3 good
**Rating:** 7
**Confidence:** 4

**Summary:**

The paper aims to fix adversarial training in graph neural networks (with respect to perturbations) by revealing limitations in prior work (from theoretical insufficiencies to experiment flaws), proposing a novel attack scheme and a superior architecture (robust diffusion) that makes adversarial training a leading defense method against such attacks.

**Strengths:**

- Paper is well written and easy to understand.
- Strong theoretical analysis supported by elegant intuition and solid experimental results.
- High-quality claim is supported by code artifacts.

**Weaknesses:**

- Readers unfamiliar with adversarial perturbations may have a hard time understanding the paper, which may warrant a short paragraph explaining the threat model for both numerical data and graph-structured data.

**Questions:**

- What is the intuition behind the "diffusion" nomenclature?

**Limitations:**

The authors have not addressed the potential negative societal impact of robust diffusion, and a short analysis in discussion may help.

---

> ### Author Rebuttal · Authors · 2023-08-09
>
> We want to thank the reviewer for the positive feedback!
>
> ## W1: Introducing adversarial perturbations
>
> >Readers unfamiliar with adversarial perturbations may have a hard time understanding the paper, which may warrant a short paragraph explaining the threat model for both numerical data and graph-structured data.
>
> We thank the reviewer for the valuable suggestion and we will address this in the revision of the paper.
>
>
> ## Q1: Graph diffusion
>
> >What is the intuition behind the "diffusion" nomenclature?
>
> A polynomial of the normalized adjacency matrix / Laplacian is generally referred to as a graph diffusion (e.g., see Eq. 1 in [T]). The most intuitive take on it is perhaps via the view of the heat kernel, which generalizes the heat equation to a discretized manifold represented by the graph. Then, the heat diffuses over the graph. We will add an explanation to make the nomenclature clear to the reader.
>
>
> ## L1: Societal impact
>
> > The authors have not addressed the potential negative societal impact of robust diffusion, and a short analysis in discussion may help.
>
> We expect the societal impact to be similar to other works on adversarial robustness, where we think the benefits outway the risks. Concretely, we allow for more fine-grained perturbation models for machine learning practitioners. Having the right tools at hand can further the reliability of graph machine learning and, thus, have a positive societal impact as long as applied ethically. Given your feedback, we will add an *Ethical Statement* to our revised manuscript.

---

> > ### Comment · Reviewer_ZTec · 2023-08-19
> >
> > Thank you for your clarifications. My score remains the same since my questions did not constitute a significant factor in a score change.

---

> > > ### Author Response · Authors · 2023-08-21
> > >
> > > We thank the reviewer for acknowledging our rebuttal!

---

### Author Rebuttal · Authors · 2023-08-09

We thank the reviewers for the constructive feedback and for acknowledging that our work is intriguing and addresses key challenges (bw8r), is insightful and useful for the community (fjD8 & zNuu), as well as provides a theoretical analysis that is supported by elegant intuition (ZTec).

We addressed all questions and important comments by the reviewers. We now include **additional experimental results**  as requested by reviewers fJD8 and zNuu (see attached pdf):

* We added **three more datasets** (one homophilic and two heterophilic).
* We added **two more attacks**, the powerful adaptive PGD attack [A] and the classic benchmark attack DICE [Q].
* We added the GRAND defense as a **new baseline defense**.

We are confident to have addressed all concerns of the reviewers and if there are remaining open questions, we are happy to engage in further discussions.

Below, we collect all references used in our rebuttal:


[A] Xu et al. "Topology attack and defense for graph neural networks: An optimization perspective.", IJCAI, 2019\
[B] Geisler et al. "Robustness of graph neural networks at scale.", NeurIPS, 2021 \
[C] Kellerer et al. "Multidimensional Knapsack Problems.", Knapsack Problems. Springer, Berlin, Heidelberg, 2004\
[D] Günnemann S. "Adversarial robustness.", Graph Neural Networks: Foundations, Frontiers, and Applications, 2022\
[E] Zügner et al. "Adversarial Attacks on Neural Networks for Graph Data.", SIGKDD, 2018\
[F] Zügner et al. "Adversarial attacks on graph neural networks via meta learning.", ICLR, 2019\
[G] Wu et al. "Adversarial examples for graph data: Deep insights into attack and defense.", IJCAI, 2019\
[H] Hussain et al. "Structack: Structure-based Adversarial Attacks on Graph Neural Networks.", HT, 2021\
[I] Zhan et al. "FHA: Fast Heuristic Attack Against Graph Convolutional Networks.", DS 2021\
[J] Yang et al. "Derivative-free optimization adversarial attacks for graph convolutional networks.", PeerJ Comput., 2021\
[K] Bojchevski et al. "Efficient Robustness Certificates for Discrete Data.", ICML, 2020\
[L] Murdoch et al. "Definitions, methods, and applications in interpretable machine learning.", PNAS, Vol. 116, No. 44, 2019\
[M] Ying et al. "GNNExplainer: Generating Explanations for Graph Neural Networks.", NeurIPS, 2019\
[N] Chapelle et al. "Semi-Supervised Learning", Adaptive Computation and Machine Learning, 2006\
[O] Xu et al. "Towards an efficient and general framework of robust training for graph neural network", ICASSP, 2020\
[P] Deng et al. "Batch Virtual Adversarial Training for Graph Convolutional Networks", arXiv:1902.09192, 2019\
[Q] Waniek et al. "Hiding individuals and communities in a social network.", Nature Human Behaviour, 2018\
[R] Wen et al. "Is Adversarial Training Really a Silver Bullet for Mitigating Data Poisoning?", ICLR, 2023\
[S] Tao et al. "IDEA: Invariant Causal Defense for Graph Adversarial Robustness", arXiv:2305.15792, 2023\
[T] Gasteiger et al. "Diffusion Improves Graph Learning", NeurIPS, 2019\
[U] Li et al. "Adversarial Attack on Large Scale Graph", TKDE, 2021\
[V] Zügner et al. "Certifiable Robustness and Robust Training for
Graph Convolutional Networks", SIGKDD, 2019\
[W] Li et al. "Spectral adversarial training for robust graph neural network.", TKDE, 2022\
[X] Mujkanovic et al. "Are Defenses for Graph Neural Networks Robust?", NeurIPS, 2022

---

### Decision · Program_Chairs · 2023-09-21

**Decision:**

Accept (poster)

**Comment:**

The reviewers appreciate the authors' contributions to elucidating an important existing limitation on the definition and setting of GNN adversarial robustness under the transductive setting. However, in order to avoid potential confusion, I would encourage the authors to better distinguish the term/notation of "algorithm" and "model". The difference in classic learning problems might be blurred, but I do believe it's worth differentiating in this paper -- $f_\theta(G)$ is supposed to be a GNN model that takes a graph $G$ as input, along with the node features, and then output the label prediction for all the nodes. Thus, the model parameter refers to $\theta$ only, and it is unclear how one can encode the clean graph $G$ observed during training into the model parameter $\theta$. In contrast, the learning algorithm can output both a model $f_\theta$ and the clean graph $G$, and then use the clean graph $G$ during inference under the transductive setting.

That being said, I share with the reviewers' opinion that this paper is a still good contribution to the community to clarify the limitation under the transductive setting, and I would encourage the authors to address the reviewers' comments and submit a revised version of the paper when preparing for the camera-ready version.